# ComplexEye: a multi-lens array microscope for high-throughput embedded immune cell migration analysis

Zülal Cibir[1,9], Jacqueline Hassel[2,9], Justin Sonneck[3,4], Lennart Kowitz[3], Alexander Beer[1], Andreas Kraus[1], Gabriel Hallekamp[2], Martin Rosenkranz[2], Pascal Raffelberg[2], Sven Olfen[2], Kamil Smilowski[2], Roman Burkard [2], Iris Helfrich [5], Ali Ata Tuz[1], Vikramjeet Singh[1], Susmita Ghosh[3], Albert Sickmann [3,6,7], Anne-Kathrin Klebl[8], Jan Eike Eickhoff[8], Bert Klebl[8], Karsten Seidl [2], Jianxu Chen [3], Anton Grabmaier [2], Reinhard Viga [2] ✉ & Matthias Gunzer [1,3] ✉

Autonomous migration is essential for the function of immune cells such as neutrophils and plays an important role in numerous diseases. The ability to routinely measure or target it would offer a wealth of clinical applications. Video microscopy of live cells is ideal for migration analysis, but cannot be performed at sufficiently high-throughput (HT). Here we introduce ComplexEye, an array microscope with 16 independent aberration-corrected glass lenses spaced at the pitch of a 96-well plate to produce high-resolution movies of migrating cells. With the system, we enable HT migration analysis of immune cells in 96- and 384-well plates with very energy-efficient performance. We demonstrate that the system can measure multiple clinical samples simultaneously. Furthermore, we screen 1000 compounds and identify 17 modifiers of migration in human neutrophils in just 4 days, a task that requires 60-times longer with a conventional video microscope. ComplexEye thus opens the field of phenotypic HT migration screens and enables routine migration analysis for the clinical setting.

Neutrophils are the most abundant leukocytes in human blood[1] and first line of cellular defense against tissue damage or invading microorganisms[2,3]. During host defense they start to invade inflamed target regions by autonomous migration just minutes after onset of inflammation[4]. However, migrating immune cells can also do harm. For example, it has been shown that the infiltration of growing tumors with neutrophils is always associated with a poor prognosis[5]. We showed,

that neutrophils massively invade human melanoma colonies in vitro and experimental tumors in mouse models[6]. Furthermore, we showed that neutrophils infiltrate other sites of sterile inflammation such as infarcted hearts[7] or stroke brains[8] and thereby aggravate the tissue damage associated with the initial ischemic insult. Hence, autonomous neutrophil migration can be a key factor for the exacerbation of cancer and systemic inflammation. But the more than a century old knowledge

[1]Institute for Experimental Immunology and Imaging, University Hospital, University of Duisburg-Essen, Essen, Germany. [2]Department of Electronic Components and Circuits, University of Duisburg-Essen, Duisburg, Germany. [3]Leibniz-Institut für Analytische Wissenschaften – ISAS – e.V., Dortmund, Germany. [4]Faculty of Computer Science, Ruhr-Universität Bochum, 44801 Bochum, Germany. [5]Department of Dermatology and Allergology, Medical Faculty of the Ludwig Maximilian University of Munich, Munich, Germany. [6]Medizinisches Proteom-Center, Ruhr-Universität Bochum, 44801 Bochum, Germany. [7]Department of Chemistry, College of Physical Sciences, University of Aberdeen, AB24 3FX Aberdeen, UK. [8]Lead Discovery Center GmbH, Dortmund, Germany. [9]These authors contributed equally: Zülal Cibir, Jacqueline Hassel. ✉e-mail: reinhard.viga@uni-due.de; Matthias.Gunzer@uk-essen.de

of neutrophil migration, which is now understood into the finest molecular details, is still not exploited to better diagnose disease states. This is astonishing considering that next to disease-promoting functions of migrating neutrophils there are also numerous examples of defective neutrophil migration being associated with different states of disease[9–13]. We also recently showed that defective neutrophil migration can herald upcoming health exacerbations several days in advance[14].

Next to diagnosing disease states by migration analysis, being able to specifically interfere with neutrophil migration would also have enormous potential for application in a clinical setting. Several receptor-ligand systems for neutrophil migration are known. Important members are the chemokine receptor CXCR2[15] and the receptors for Leukotriene B4 (BLT2)[16] and especially those for formylated peptides (FPR)[17]. The bacterial peptide n-formyl-l-methionyl-l-leucyl-l-phenylalanine (fMLP) is one of the strongest chemokinetic and chemotactive substances that after binding to formyl-peptide receptors (FPR1 & FPR2) on neutrophils induces strong cell migration[17,18]. While migration following fMLP gradients leads neutrophils into infectious foci to trigger anti-infectious responses[19], it can also steer the cells into sites of sterile inflammation based on the release of formylated peptides from mitochondria of dying host cells[20]. This can have "healing" consequences, e.g., due to increased re-vascularization of necrotic tissue[21], but can also severely exacerbate a pathological state such as after stroke[8] or in most cancers[5,22]. Given the very different sources of formylated peptides (somatic cells vs. pathogens) and the associated environmental conditions of sterile and infectious inflammation it is very likely, that the molecular mechanisms underlying the response of neutrophils to these conditions are different. It might be possible to generate selective modulators, that interfere with one while leaving the other response unchanged and hence allow for example the inhibition of neutrophil migration into tumors without blocking general neutrophil immune responses. However, since the number of ligands triggering FPR1 or 2 is very high and molecularly diverse[18], it is not straightforward to predict modulatory functions for compounds by e.g., modeling. Instead, a high-throughput (HT) screen of compound libraries for migration modifiers would be required.

Physiologically, cell migration occurs in intact organs where it is studied with advanced intravital microscopy[23,24]. However, intravital microscopy is not suitable for HT analyses and also unavailable for human cells. Instead, isolated cells can be studied in appropriate assays in vitro[24]. The most efficient approach hereby is time-lapse video microscopy of hundreds of individual cells with sub-cellular resolution combined with automated cell tracking. Only this approach can document and quantify all features that are essential to comprehensively capture cell migration. These include permanently changing shapes of cells, active and non-active cells in a group, chemotaxis versus chemokinesis or frequency and duration of cell-cell contacts. Immune cells also transiently pause and then resume migration in a burst[25]. Migration inducers can thus act by reducing pauses which leads to enhanced group motility in a process called recruitment[17]. Recruitment can occur in the absence of directed migration and hence is difficult to discover with assays that specifically focus on chemotaxis[10]. The advantages of video microscopy for cell migration studies are, therefore, widely recognized among experts. But although today the analysis of such videos can be performed by automated cell tracking[26], the production of sufficient numbers of movies is a bottleneck, since no HT video microscopes for fast moving cells are available. Therefore, the majority of immune cell migration studies relies on endpoint methods, that quantify the migration of entire populations at large (i.e., lacking single-cell resolution) after pre-set time points and only analyze the cells that have traveled the farthest[10]. Although much easier to perform and analyze compared to live-cell video recordings, endpoint methods cannot capture and evaluate many essential parameters of cell migration. Thus, there is an obvious lack of HT video microscopy for migration studies. This impedes the enormous potential of routinely measuring the migration of immune cells as a diagnostic tool for multiple diseases in the clinics or to find novel migration modifiers.

The most common format for HT microscopy is the 96-well plate or its high-density variants (384 or 1536 wells on the same footprint)[27]. Since a useful HT video microscope must be able to record images in 96-well plates, present-day commercial video microscopes typically feature a single lens and a motorized XYZ-stage to record frames at a given interval and move the plate from well to well in between. While suitable for tumor cells, where recording one frame per 30 min is sufficient to capture their movements[28], this approach fails with immune cells. Immune cells are ~20–100 times faster[11,12] and much smaller than tumor cells. It thus requires one frame at least every 8 s to allow effective tracking of fMLP-stimulated human neutrophils[11,29]. Since too high travel speed of the stage induces shaking artifacts in non-adherent cells, a single-lens system can only monitor four wells at the speed required for neutrophil cell tracking. The solution would be microscopes based on multi-lens setups. Indeed, experimental systems have organized 12 conventional 40x lenses into an array[30]. This 12-channel microscope features an electrical XY-stage for the movement of the 96-well plate against a stable set of 12 objectives. In that sense it is not different from a standard single lens system with an electrical stage and moving the stage too quickly would also cause movement artefacts. Therefore, the 12-channel system of Cribb et al. could image maximum 48 wells of a 96-well plate per 8 s to study non-adherent cells. In general, microscopes that move the plate quickly reach limits of imaging non-adherent cells. The recent description of a 96-eye plastic-mold lens system goes into the right direction, but due to insufficient optical resolution requires excessive post-processing, limiting recording speeds to one frame per 90 s in 96 wells[31].

Here we present our microscope, the ComplexEye. ComplexEye is a multi-lens video microscope, that employs an array of 16 aberration-corrected glass lenses, each with image detector and individual illumination, to simultaneously image 16 adjacent wells of a 96-well plate or 64 wells of a 384-well plate with one frame per 8 s. ComplexEye is different by design, as it moves the optics against a stable plate. Hence, stage travel speed can be increased to boost the throughput of the system without having to fear movement artefacts in non-adherent cells. ComplexEye is 16 times faster than conventional time-lapse microscopes yet with comparable resolution and video quality. In this work, we simultaneously measure neutrophil migration of up to 16 volunteers in a standardized clinical routine-ready assay. In addition, we rapidly screen 1000 compounds for their ability to modify fMLP-induced neutrophil migration and identify more than 15 substances with inhibitory function on various levels of motility.

## Results

### ComplexEye concept and design

The neologism "ComplexEye" is derived from the German word "Komplexauge," which describes the compound eyes of arthropods. They consist of hundreds to thousands of individual imaging subunits called ommatidia, which together form a high-performance device for object and motion detection[32]. ComplexEye is characterized by four main features. First, to enable parallel acquisition of as many independent cell samples as possible in a multi-well plate the system was designed with currently 16 and can potentially host 96 individual microscope systems consisting of lens, illumination, and imaging unit in the pitch of a standard 96-well plate. A key challenge was the lack of off-the-shelf lenses that would fulfill the optical requirements (>4x magnification, NA 0.3) while having an outer diameter of just 8 mm to fit under single wells of 96-well plates (Fig. 1a). Available microscope objectives are typically 25–35 mm wide, which is why until now multi-lens microscopes just used staggered arrays of e.g., twelve[30]. Hence, we developed a proprietary solution. Based on own physical designs a lens maker produced brass cylinders with 8 mm outer diameter containing

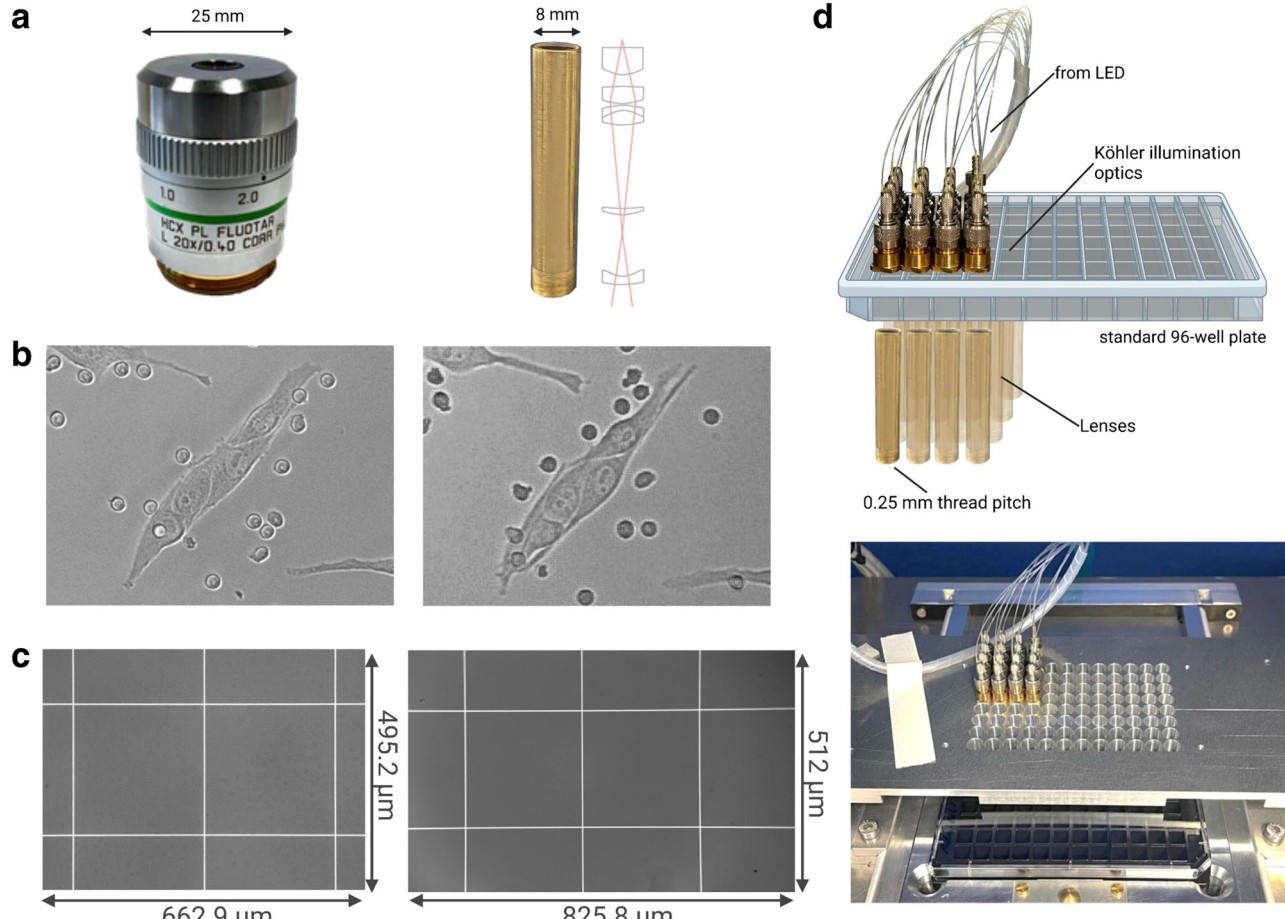

**Fig. 1 | Optical performance of the ComplexEye lens. a** Left: Photograph of a standard 20x microscope lens. Right: The ComplexEye lens with its schematic optical path. The dimensions above show the difference in diameter of the lenses. **b** Images acquired with a standard 20x lens (left) and just minutes later with the ComplexEye lens (right) of the identical cell sample consisting of Cutaneous Melanoma (CM cells) and neutrophil granulocytes. This experiment was repeated three times with similar results. **c** Images acquired with a standard 20x lens (left) and the ComplexEye lens (right) of a 250 μm grid in a Neubauer chamber. This experiment was repeated twice with similar results. The size of the FOV in the respective captured images are displayed. **d** Above: Schematic structure of the 16 individual microscope units without the imagers. Below: Photograph of the top view of the ComplexEye in its working setup. LED Light Emitting Diode.

6 glass elements that are exactly positioned inside of the cylinders. Connected to a megapixel sensor these lenses generate images that are optically comparable to those made with a conventional microscope (Fig. 1b, Supplementary Fig. 1) and, unlike images in a recently published array microscope, do not require computation-intense postprocessing[31]. This was verified by imaging USAF Patterns to compare the optical resolution in different parts of the field of view (FOV) of the ComplexEye and a commercial microscope (Supplementary Fig. 1). The optical resolution of our system is exclusively defined by the NA of the optics, which is 0.3 for ComplexEye and thus leads to 917 nm at 550 nm illumination. In the Leica system we used a 20x lens with a NA of 0.4, hence an optical resolution of 688 nm. Next, a Neubauer counting chamber was used to acquire images with a conventional microscope and our array system to determine the total FOV. (Fig. 1c). Thereby, the conventional Leica system demonstrated a FOV of 662.9 × 495.2 μm and the ComplexEye 825.8 × 512 μm. The magnification of the ComplexEye optics is thus 4.69, while for the Leica this is 13.43. In a suitable array microscope each lens requires its own Köhler-optimized illumination[33] to achieve perfect image quality. We solved this problem with a white LED whose light was distributed by polymer fibers to each of 16 focusing elements, generating a homogenous illumination (Fig. 1d, Supplementary Fig. 2). Thereby, the power of illumination related to the illuminated area (0.031 cm²) was determined as 30 nW.

To obtain a working system, we developed a custom-built circuit board containing a Spartan 6 Field-programmable Gate Array (FPGA) from Xilinx to control imager functionality and data transfer to a control PC. We positioned the FPGA outside of the imager area, which was essential to allow the necessary tight packaging of the imaging CMOS chips (Fig. 2a). The second key design feature of ComplexEye was the ability to move the optics and lighting around a stationary well stage (microscope moves around resting sample), thereby avoiding movement artifacts in the biological sample, which is essential when imaging non-adherent immune cells. This was achieved by rigidly connecting optics and illumination and mounting the entire ensemble on a motorized XYZ-stage (Fig. 2b, Supplementary movies 1 and 2). Thereby, the observed position of all microscope units can be simultaneously changed without moving the multi-well plate. ComplexEye features illumination times of 1/30 s, XY-travel speeds of 10 mm/s and 20 mm/s of the Z-drive for focusing. One entire process of positioning, focusing and imaging is done in under 2 s allowing to visit four positions per lens, before the first position has to be imaged again (Supplementary Fig. 3). The rigid combination of optics and illumination was also a prerequisite to image 384-well plates. For this, the entire array of 16 microscopes only had to move in a small square (movements of 4.5 mm in the X or Y direction shift the focus from the center of one well in a 384-well plate to the next). With only 4 such movements, 64 wells could be imaged within 8 s thereby generating series of

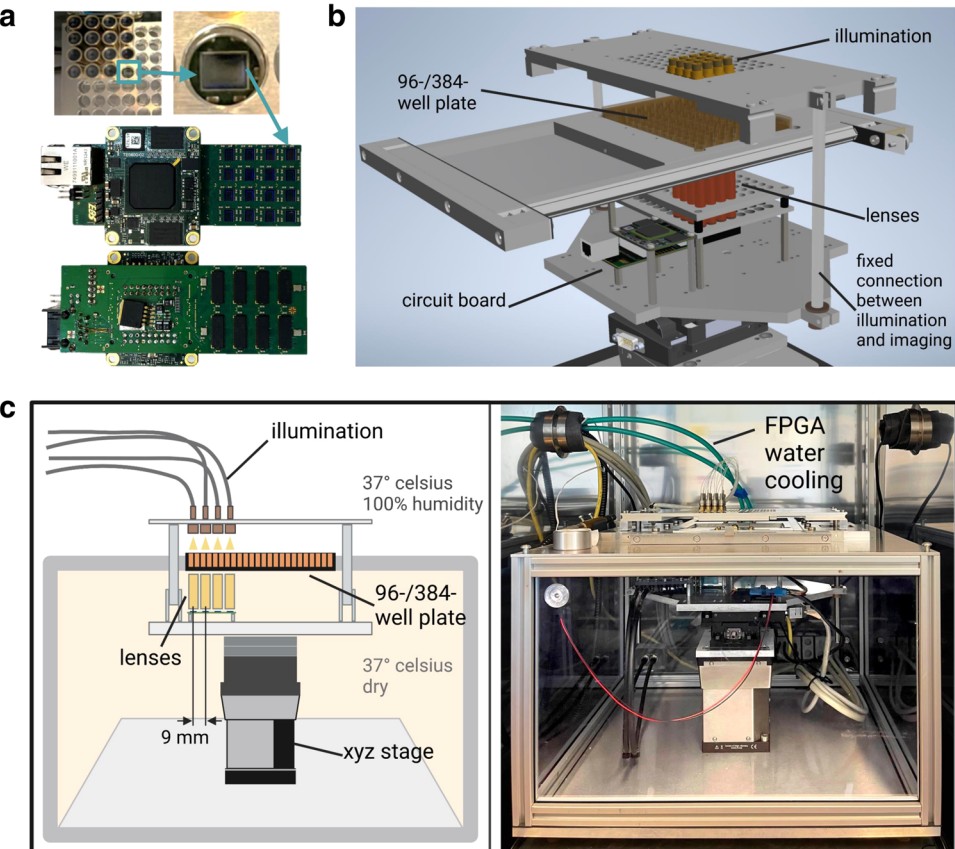

**Fig. 2 | The ComplexEye: a multi-lens microscope for high-throughput, embedded cell migration analysis. a** Above: Top view of the lenses with one lens removed to allow a zoom onto the underlying CMOS-imager. Below: photographs of the circuit board with imagers and FPGA from top and bottom. **b** 3D model of the ComplexEye excluding the dividing chamber (see also supplementary movies 1 and 2). **c** Left: Schematic overview of the ComplexEye components and structure inside the tempered cabin with the division into dry and humid climate zones. Right: Photograph of the real-world ComplexEye corresponding to the view shown in the schematic figure (left). FPGA Field Programmable Gate Array.

64 independent movies with the required time resolution, each visualizing hundreds of migrating immune cells at sub-micrometer resolution.

Conventional cell observation microscopes often suffer from focus drifts induced by uneven heating of the entire system. To avoid this, placing the entire system into a standard cell-culture-grade tempered cabinet, was the third design principle. Thereby, the ambient temperature could be closely monitored, and optimal conditions could be ensured (Fig. 2c). This construction demonstrated its performance by allowing to generate hour-long video sequences of slow or fast migrating cells without any focus shift or movements of the FOV (Supplementary Movies 3 and 4). To protect the electronics from the humidity in the incubator, we encapsulated them to separate them from the sample level (Fig. 2c). Due to the substantial heat emission of the FPGA and the imaging-related components we had to construct a liquid cooling system, that regulates the temperature of the electronic components during image acquisition and avoids convective heating of the cell-cultures (Fig. 2c).

As a result of unavoidable tolerances in manufacturing and assembly of components and assemblies of the array microscope core unit and the well plates, individual focus differences between the wells remained. Therefore, the fourth key feature of ComplexEye was collective focusing to ensure that all generated movies were suitable for later analysis. To achieve this, the focus position map-based triggered mode was used (see methods and Supplementary Fig. 3 for detailed information). Here, the focal plane was calibrated at the beginning of each measurement for all 16 lenses. Then, during acquisition, the whole 16-fold lens unit was moved periodically up and down within the

overall focus range while in every round each of the 16 imagers took a picture in the moment of maximum image sharpness at the predefined frame to frame time-gap of the resulting movie.

## Migration measurements and comparison with a conventional microscope

To benchmark the ComplexEye system we employed a previously established clinical routine-ready migration assay for human neutrophils from peripheral blood[11,14]. Briefly, neutrophils were isolated from peripheral blood of healthy volunteers, plated in 16 adjacent wells of 96-well plates and treated with PBS as a control or fMLP, CXCL1 or CXCL8. CXCL1 and CXCL8 belong to the CXC chemokine family and act as chemoattractants for neutrophils[18]. The wells were imaged with ComplexEye at a rate of one frame/8 s for one hour. The generated time-lapse movies were automatically tracked and the results compared. As the ComplexEye was able to simultaneously image 16 wells of a 96-well plate, we used quadruplets of all conditions (Fig. 3a). The analysis showed that the measured values were comparable to our migration data obtained before[14] (shown as orange lines in Fig. 3a). Furthermore, the four repetitive measurements of each condition showed the high reproducibility of migration values obtained with the ComplexEye system (Fig. 3a). In addition, we compared the performance of the ComplexEye with a standard cell migration microscope (Leica DMI6000 B with motorized stage). For this we measured neutrophil motility of cells from the same preparation simultaneously under the exact same conditions at the ComplexEye and the Leica system (Fig. 3a). The tracking results and also the quality of the movies were comparable (Fig. 3a, Supplementary Movie 5). Next, we tested the

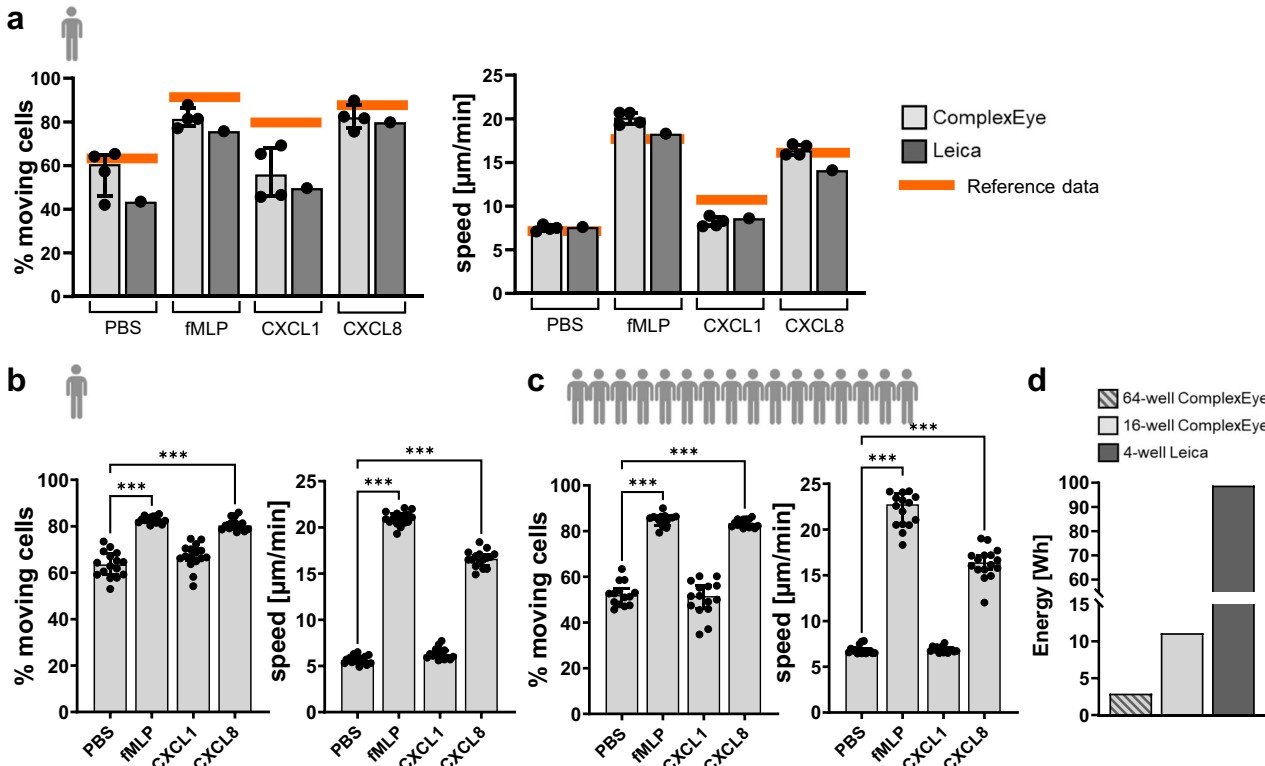

**Fig. 3 | ComplexEye performance and comparison with a conventional microscope. a** 16-well measurement of freshly prepared human peripheral blood neutrophils from a single donor ($n = 1$) migrating in response to the indicated stimuli. Each condition was prepared in quadruplicates in independent wells of a 96-well plate and imaged for one hour (8 s between frames) with ComplexEye using a 96-well plate. The same cells were measured simultaneously on a commercial video microscope from Leica. Shown are the tracking results of the four independent runs on ComplexEye as dots (light gray bars) and the parallel run on the Leica as single bar with dot (dark gray bars). Orange lines indicate published reference data of $n = 25$ healthy individuals[14]. **b** 64-well recording of neutrophils (one hour, 8 s between frames) from a single donor ($n = 1$) on ComplexEye using a 384-well plate with the indicated stimuli in 16-fold repetitions. **c** 64-well measurement as in (**b**) but with cells simultaneously prepared from 16 donors ($n = 16$). Each cell preparation was measured with four different conditions as indicated. **d** Comparison of energy consumption between ComplexEye and a commercial system from Leica for the generation of a one-hour movie with 8 s between frames. The striped bar demonstrates the 64-well measurement at the ComplexEye, the light gray bar the 16-well measurement at the ComplexEye and the dark gray bar the 4-well measurement at the Leica. Statistical significances were calculated via Kruskal–Wallis test with multiple comparisons. Data are presented as median values ± interquartile range. ***$p < 0.001$. Source data are provided as a Source Data file.

high-density well capacity of ComplexEye, since the 16-lens ComplexEye was able to simultaneously record movie sequences in $4 \times 16$ (=64) wells of a 384-well plate. First, we investigated neutrophils from a single donor in 64 wells with the same four different conditions as above, yet in 16-fold repetition (Fig. 3b). The data showed a very high homogeneity and results comparable to those of the 16-well measurements and our previous data[14]. Next, to estimate the spread of results obtained with various donors and also to demonstrate the ability to run many independent samples in one go we measured the motility of neutrophils isolated simultaneously from 16 healthy donors (Fig. 3c). Here we again revealed tracking results comparable to our database, yet displaying the spread of results seen with many individuals[14]. Finally, in times of high energy prices any novel HT assay must address the problem of energy consumption to not become inhibitively expensive. Hence, we compared the total energy usage of all components (including the cell incubator) of the 16- and 64-well setups of the ComplexEye and the conventional microscope to generate one-hour movies with a time resolution of one frame every 8 s. We measured an aggregated 2.9 watts/movie for the 64-well setup and 11.1 watts/movie for the 16-well setup, while a similar movie generated on the commercial microscope required 98.8 watts. Hence, the ComplexEye is over 34-fold more energy-efficient than conventional systems (Fig. 3d). Furthermore, besides measuring chemokinetic migration, ComplexEye is also able to investigate chemotaxis. In a bead-based assay we also discovered a very unusual behavior of

human neutrophils when encountering fMLP on a solid source as opposed to its availability only as a soluble factor (Supplementary Figs. 4 and 5 and Supplementary Movie 6).

## High-throughput screening of 1000 compounds
As laid out in the introduction, ComplexEye is intended to make immune cell migration a cellular behavior that can be read out in HT fashion for diagnostic purposes (Fig. 3a–c) but is also applicable in novel phenotypic compound screens. To demonstrate that this is indeed possible, we searched for modifiers of fMLP-induced migration in human neutrophils. For this we screened a library of 1000 known bioactive compounds and tested their ability to interfere with the motility of freshly isolated human neutrophils migrating in response to fMLP (Fig. 4). Blood neutrophils were isolated from healthy volunteers, plated in 64 wells of a 384-well plate, treated with one of the 1000 compounds and stimulated with fMLP. The plates were imaged with a rate of one frame/8 s for 1 h at 37 °C. The generated time-lapse movies were then automatically tracked and analyzed (Fig. 4a). ComplexEye allowed to simultaneously screen 61 compounds together with three controls (PBS, DMSO and fMLP or fMLP/DMSO alone) in one round. With this setup we were able to screen all 1000 compounds in only 4 days and with 17 single rounds using cells from 8 volunteers (Fig. 4b). In a conventional single-lens system the same screen would have required 1000 rounds and taken almost 60 times longer. The detailed tracking results of all tested compounds and the appropriate controls

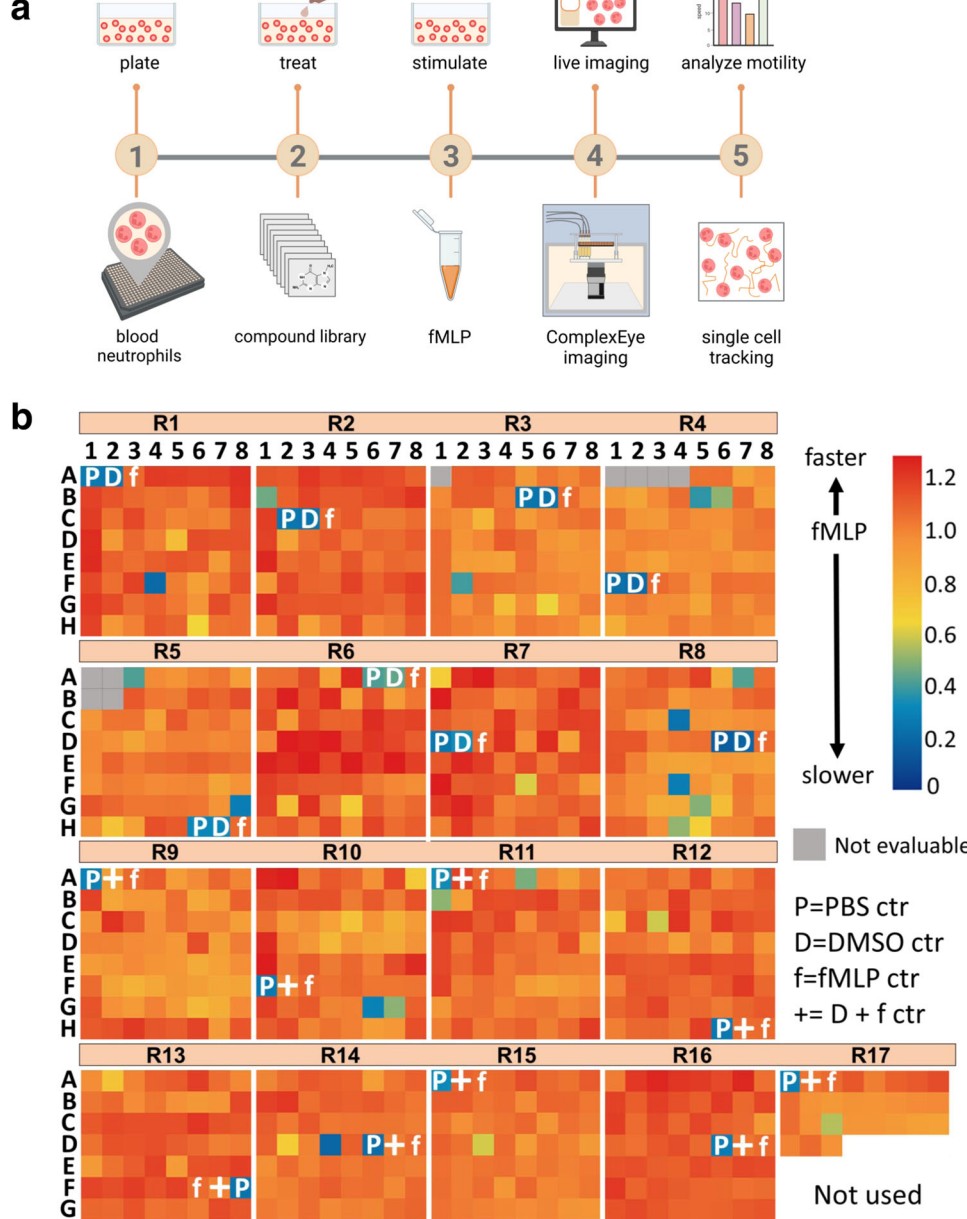

**Fig. 4 | ComplexEye high-throughput screening of migration modifying compounds. a** Experimental setup of the screening assay. Briefly, neutrophils from human blood were isolated and plated on a 384-well plate, treated with one of the 1000 compounds from a library of known bioactives and stimulated with fMLP. Neutrophil motility was then recorded simultaneously in 64 wells of a 384-well plate for one hour (8 s between frames) using ComplexEye. Afterwards the motility was analyzed via single cell tracking. **b** Data represent 1000 movies, ~800 tracks/movie

and show the impact of 1000 compounds screened in 17 rounds, each round with three controls (PBS, DMSO and fMLP or fMLP/DMSO). The heatmap shows each round with 64-wells with the relative speed of imaged neutrophils indicated as color code compared to the fMLP-control in that run (artificially set to 1.0). Compounds that reduced the speed are shown in green-blue (low speed). Indicated gray wells were non-evaluable due to production residues of the 384-well plates inhibiting clear sight of the cells. Source data are provided as a Source Data file.

are summarized in the supplementary files (Supplementary Data 1). In total we tracked more than 1.2 million individual neutrophils (mean 1245 cells per movie) for at least one minute (8 frames) or longer in a total of 1042 movies. This ensures a very robust data base for our analysis.

Single cell tracking revealed 17 compounds that reduced fMLP-induced neutrophil motility by more than 40% (Fig. 5a, Supplementary Movie 7, Supplementary Table 1) while additional compounds still had a reducing capacity of 20–40% (Supplementary Fig. 6). Furthermore, 8 compounds increasing the fMLP-induced speed by more than 20% could be identified (Supplementary Table 2). Interestingly, the

parameter "relative activity", describing the number of migrating cells relative to the fMLP control was also reduced by 20 up to 90% by some compounds (Fig. 5b, Supplementary Movie 8, Supplementary fig. 6). Thereby the effect of speed- or activity-reduction could be separated in various compounds (Fig. 5c), while other compounds modified both parameters simultaneously. This allowed to sort compounds into clusters of action (Fig. 5c). Importantly, some of the speed-decreasing compounds also had an effect on neutrophil morphology. In our migration assay, control neutrophils treated with fMLP were characterized by low adhesion and a compact migratory shape[11] (Fig. 5d). In contrast other compounds clearly changed the morphology of the

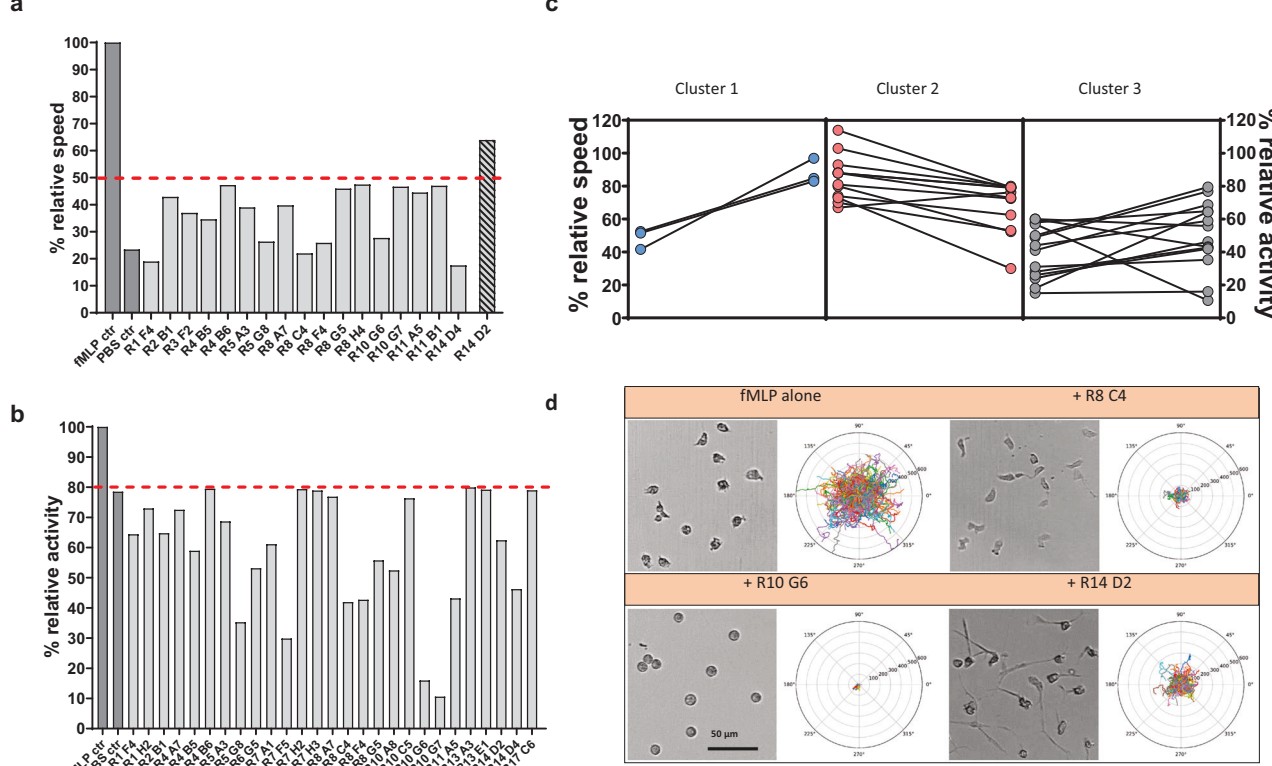

**Fig. 5 | ComplexEye high-throughput screening identifies neutrophil migration modifiers. a** Relative speed data with 17 compounds reducing motility by more than 40% compared to fMLP as detected by the screen illustrated in Fig. 4. Compound R14D2 had less effects on speed, but strongly affected the cell shape. **b** Relative activity data with 27 compounds reducing the number of migrating neutrophils by more than 20% compared to fMLP. **c** Sorting of inhibitory compounds into classes according to their effect on speed and activity of migrating neutrophils. In every square the left vertical line is for relative speed and the right vertical line is for relative activity. Class 1: compounds strongly decreasing the speed whereas the number of migrating cells was not affected (lines with blue dots).

Class 2: compounds strongly decreasing the number of moving cells without affecting their speed (lines with red dots). Class 3: compounds decreasing both, speed and activity (lines with gray dots). **d** Comparison of neutrophil morphology between fMLP-treated cells and cells treated with fMLP and the indicated compounds. This experiment was not repeated, Polar plots show migration tracks of all cells in the experiment normalized to one common center. Rings in polar plots define 100 μm distances. The scale bar is given as 50 μm. All datasets shown in this figure are based on cells from a total of eight single donors analyzed within 4 consecutive days. Each condition was measured once ($n = 1$). Source data are provided as a Source Data file.

neutrophils, while having distinct effects on the extent of migration, as indicated in polar plots of migration paths (Fig. 5d, Supplementary Movie 9). In addition, we identified compounds with comparably small effects on neutrophil motility but a very strong impact on cell morphology like e.g., compound R14D2 (Fig. 5a, d). Neutrophils treated with R14D2 were abnormally shaped, more flattened and presented with an elongated structure due to difficulties to de-adhere the cellular uropod (Fig. 5d, Supplementary Movie 9). Also, the reaction of neutrophils towards soluble versus bead-bound fMLP were remarkable (Supplementary Fig. 5). This highlights the importance of visual cell shape analyses for HT immune cell migration screens. In summary, with the help of a ComplexEye HT-screen we identified many compounds with the previously unknown capability to modify neutrophil migration and shape. Of the 17 compounds identified in the first screen 12 (71%) could be validated for their function in a second analysis (Supplementary Fig. 7). Future studies will now allow to investigate the underlying cellular mechanisms in more detail and hence arrive at a deeper understanding of fMLP-induced neutrophil motility.

## Discussion

With ComplexEye we introduce a new kind of array microscope that is able to deliver the image quality and framerate required to make immune cell migration a parameter suitable for HT screens. This is an important achievement since the fact that neutrophils are able to migrate is known for more than 150 years[24] but apart from some

interesting pilot studies[34] that knowledge is not exploited beyond detailed investigations into the underlying cell biological mechanisms. With our screen we now identify a whole group of substances that can modify fMLP-induced neutrophil motility and hence allow to study the effect of migration modification itself in suitable model systems, e.g., of cancer growth or infection control in preparation for future clinical studies. It is self-evident that similar screens can be performed with other neutrophil migration inducers such as C5a or LTB4[18]. Such data form the basis for any later application of neutrophil migration modifiers in clinical settings, e.g., to stop neutrophil infiltration of tumors or other sites of sterile inflammation[24]. Beyond therapeutic applications measuring immune cell migration has a great potential as diagnostic tool or to detect early warning signs of disease exacerbation. For instance, reduced neutrophil migration underlies increased infection susceptibility[35] and in the prodromal phase of sepsis neutrophil migration changes long before other clinical signals take effect[36]. In a limited clinical trial also we observed that neutrophil motility in a standardized in vitro assay was reduced in liver cirrhosis patients several days before they developed sepsis or other disease exacerbations[14]. We also demonstrated that the severity of myelodysplastic syndromes was closely associated with the degree of migration depression in patient´s neutrophils[11] and migration defects were a hallmark of chronic myeloid leukemia[37] and atypical chronic myeloid leukemia[12]. Taken together these examples highlight the diagnostic potential of migration studies.

ComplexEye analyses are not limited to the investigation of neutrophils. We have seen that also tumor cells (Supplementary Movie 3) migrate very well in this assay and we think that analyzing the migration of other immune cells such as T cells, Monocytes and Eosinophils will also be possible after the necessary adaptations of the conditions. With ComplexEye it is, therefore, possible to rapidly obtain migration analyses from all immune cell subtypes and identify novel migration modifiers. This enables HT phenotypic screens[38] with (immune) cell migration as readout.

Interestingly, the compounds we have identified as neutrophil migration modifiers belong to a very heterogenous group of substances known to affect a very diverse set of signaling modules (Supplementary Table 1 and 2). Among these compounds are different kinase inhibitors e.g., targeting phosphatidylinositol 3-kinase (PI3K), protein kinase C (PKC) and cyclin-dependent kinase. Others target reactive oxygen species, ferroptosis, apoptosis and autophagy. Some compounds not only decreased neutrophil speed but also significantly changed their morphology. Moreover, there were analogous compounds with the same chemical background and the same target, that did not have effects on neutrophil motility. Such compounds are very valuable tools to get a deeper understanding of the cytoskeletal rearrangements and the interaction of leukocytes with their environment when migrating[39]. None of these compound effects could have been predicted from structural or functional data and hence required a migration screen to be identified. ComplexEye represents the tool to perform much more extensive studies of that field in the future. Available chemical libraries can contain several 100,000 or even millions of compounds with completely unknown function[40,41], that can now be screened.

In its current setup the performance of ComplexEye is not only comparable to other microscopes in terms of data quality, but at the same time much more efficient in data generation and hence also very energy-efficient. Similar to the green-IT concept[42] this opens the field of green microscopy, while not compromising but rather enhancing performance. In contrast, a recently introduced microscopy system with 96 plastic-mold lenses is limited to 96 wells and requires immense computing power to reach the desired image quality. With the turn-around time of one frame per 1.5 min it is also far too slow to analyze fast moving cells like immune cells[31]. We demonstrate instead, that the use of optimized aberration-corrected glass lenses is critical to obtain perfect image quality from the start, an indispensable prerequisite for the high imaging speed required for useful HT migration screens.

The system shown here is only the starting point for additional developments to increase the performance of ComplexEye and thereby further open the field of HT cell migration analysis. Crucial steps will be the expansion from 16 to 96 lenses to exploit the maximum capacity of the system. This is not trivial as it requires the integration of 6 times as many optical and electronic components with the associated problems regarding FPGA communication and management of much higher video bandwidth. However, although challenging to realize we do not see fundamental roadblocks since currently the FPGA only handles the autofocus and image acquisition. This does not max out its computing power which is why on-the-fly, embedded cell segmentation will also be one of the main future goals. Then, no expensive and energy consuming CPUs and GPUs will be needed during the entire workflow of the system, further reducing the energy consumption and increasing throughput of the complete system.

Currently, ComplexEye only enables the acquisition of bright-field images. This is sufficient to study purified immune cells of one sort[11] but it is unable to e.g., visualize intracellular signaling processes or distinguish different types of cells, if these have similar outer shapes. For this, fluorescence microscopy is highly useful[43]. Fluorescence microscopy requires defined excitation light as well as suitable filters separating the excitation from the emission of fluorophores[44]. Especially in live-cell imaging, it is also essential to allow rapid switching of the excitation to avoid bleaching. The realization of fluorescence is straight-forward in conventional one-lens systems, where the illumination beam path is oriented at 90° relative to detection. In a 96-well array microscope, the setup is physically not possible hence needing solutions with highly optimized multi band-pass filters in front of the detectors. Ideally, ComplexEye should be equipped with fluorescence capability without having to change the system, but this requires additional sophisticated engineering and hence is beyond the current study.

In summary, we show here an array microscope with unique capabilities for HT video-rate imaging. We have just started to scratch the surface of projects that are possible with its help. Future studies should exploit the capabilities of the technology also for fields beyond immune cell migration such as viral neutralization assays[45] or antibiotic susceptibility screens with bacteria[46] thus enabling new ways for diagnosis and discovery in a clinical setting.

Although ComplexEye is still a purely experimental system in its current state, we would nevertheless like to provide some rough estimates for its costs. These do not consider price fluctuations or costs for personnel, development or assembly. The procurement costs of all components, semi-finished products and functional assemblies for the 16x system amount to about 50,000 €. The largest items here are the lenses at around 12,000 € and the high-precision stage system at 18,000 €. For a 96x system, which would include 6 of the 16x camera clusters shown in this study, pure hardware costs of about 100,000 € can be expected at current prices.

## Methods

### Blood samples and participant information

Samples of blood in EDTA-supplemented tubes from healthy volunteers were collected by medical professionals in our faculty. The study was approved by the institutional ethics committee of the medical faculty of the University Duisburg-Essen (internal number 21-10184-BO). The studies followed strict internal and external quality assurance protocols. For our studies we included in total $n = 26$ volunteers. For the experiment in Fig. 3a, neutrophils from a 54-year-old male donor and for the experiments in Fig. 3b from a 28-year-old male donor were collected. For the experiment in Fig. 3c, neutrophils from 11 female and 5 male donors between 20 and 54 years were collected. For the compound screening experiments shown in Figs. 4 and 5, 3 female and 5 male donors between 25 and 54 years were used. However, age and sex were not relevant for our studies. All volunteers provided informed consent and were not financially compensated for their participation in our study.

### ComplexEye design, manufacturing and assembly

System Design: The mechanical components of the ComplexEye system were designed using a 3D CAD program (Inventor 2019; Autodesk, Dublin, Ireland). A PCB design program (Eagle 7, CADSoft, München, Germany) was used for circuit board design. An integrated development environment (ISE Design Suite, Xilinx, San Jose, USA) was used for structure description and programming of the FPGA in the VERILOG description language.

The basic housing, which accommodates all the moving electromechanical, optical, fluidic and electronic components of the microscope's core assemblies, is built up as a frame structure from cut-to-size strut profiles (Bosch-Rexroth 20 × 20 mm, Lohr am Main, Germany) and using connecting, sealing and terminating elements provided as part of the assembly program. The profile frames of the side walls are closed with transparent plate elements made of PMMA (thickness 5 mm), the lid and bottom are closed using plate elements made of aluminum (thickness 3 mm). To ensure the hermetic integrity of the enclosure, all connecting cables and coolant hoses are fed in at the rear of the enclosure via a polyamide cable entry panel (KEL-Quick 24/10, Icotek, Eschach, Germany) and associated sealing profile inserts.

A cut-out in the cover plate, in conjunction with a drawer system constructed from CNC-machined fittings made of aluminum and stainless steel, provides an airlock for inserting and replacing the well plates, preventing the exchange of moist air outside and dry air in the housing chamber. At the joint between the drawer and the housing cover, 1 mm thick PTFE strips act as sealing and sliding elements. The well plate itself is part of the sealing concept. For this purpose, a suitably cut rubber seal is inserted in a groove of the plate receptacle in the drawer construction made of styrene-butadiene rubber roll material. A manually unlockable, sliding spring mechanism allows the well plate to be changed easily and at the same time ensures that the plate is pressed firmly and evenly into the rubber bed of the seal.

The core assembly inside the basic housing is the multi-video microscope unit, which can be moved by motors in three axes and is constructed in a stacked design. The basic element is a precision lifting stage (L-310, PI miCos GmbH, Eschach, Germany) as a Z-platform (travel 26 mm, step resolution 0.2 μm, positioning speed 20 mm/s, lifting capacity 5.5 kg), which is screwed to the base plate of the basic housing. Two linear stages (MTS-65, PI miCos GmbH), mounted rotated by 90° in the plane, are attached to this via an adapter plate building a X-Y platform (travel 26 mm, resolution 0.1 μm, positioning speed 10 mm/s). The upper of the two linear stages carries the load-bearing aluminum base plate (thickness 10 mm) to which all subcomponents of the multi-video microscope unit are attached and which is moved as a unit by the X-Y-Z platform.

The video microscope unit consists of a proprietary circuit board (8-fold multilayer, FR4), on the upper side of which $4 \times 4$ image sensor chips (OV9712, OmniVision, Puchheim, Germany) are arranged in a square in a well grid (center distance to each side 9 mm) as well as components for their control and on the lower side of which multiplexers for canalizing the image data transfer are located in the same number below the image sensors. Via two connectors ($2 \times 100$ pin connectors) this board is connected to an FPGA board (TE0600 with Xilinx Spartan 6, Trenz-Elektronic, Hüllhorst, Germany) and in combination with this board forms the cluster controller. The FPGA board has $2 \times 512$ MByte DDR3 SDRAM with 125 MHz clock rate for temporary image data storage and a 10/100/1000 (Gigabit) Ethernet transceiver (PHY) for image data transmission to a control PC. The Cluster Controller is mounted as a unit to the base plate via spacer bolts. Heat exchangers (Twinplex), coolers, pump, control and auxiliary components (Aqua Computer GmbH & Co. KG, Gleichen, Germany) are used to cool all active components with high energy consumption such as image sensors and FPGA via a liquid cooling system. Only the heat exchangers with direct contact to the heat sources are located inside the base housing, which are connected to the other external components via coolant lines. This prevents convective heat transfer from the electronic components into the well plate.

On top of the image sensor array, also fixed via spacer bolts, is the lens carrier with threaded holes in the $4 \times 4$ grid of the image sensors, into which proprietary cylindrical mini objectives with a diameter of 8 mm are screwed. The lenses, which are manufactured to design specifications by Qioptiq Photonics GmbH & Co. KG (Göttingen, Germany), are stacked and screwed together building an aberration-corrected 6-lens system within a brass cylindrical shell and have a numerical aperture of 0.3 at a magnification of 8x.

Situated above the well plate and thus outside the base housing is the lighting unit as a bridge construction. Its base body consists of a carrier plate (10 mm thick), CNC-machined fittings made of aluminum, and cylindrical sliding elements made of PTFE (1 mm thick) as feet. The carrier plate has cylindrical fitting holes in the grid of the well plate arrangement (hole spacing 9 mm × 9 mm) to accommodate the Köhler optics. Light from a central illumination source external to the housing is distributed via plastic optical fibers and coupled per well via an optical connector (SMA) into a proprietary optical imaging system (Collischon Optic Design, Erlangen, Germany based on 2 plano-convex lenses from Edmund Optics, Barrington, USA) with Köhler characteristics (Köhler optics) and conditioned for well illumination. On sliding feet, the illumination unit stands displaceably above the well plate on the cover plate of the base housing. Two additional square cut-outs in the cover plate, in conjunction with a parallel rod guide, allow the illumination unit to be moved synchronously along with the multi-video microscope unit, which is located inside the base housing and can be moved in three axes. The guide rods are movably connected to the base plate of the multi-video microscope unit via oil-soaked, maintenance-free sintered bronze plain bearings (flange sleeve to DIN 1850V, Øᵢ 8 mm, length 16 mm). To ensure the sealing of the housing in the lead-through area of the guide rods, bellows (type FB/10/64-8-50 K, Industriebedarf Grafe, Limbach-Oberfrohna, Germany) are used.

The basic housing is located completely in the chamber of an incubator (Heratherm IGS 180, Thermo Scientific, Dreieich, Germany) for maintaining the ambient climate (37 °C). For decoupling against vibration and impact sound at the place of installation, both the incubator itself against the floor and the basic housing against the incubator are mounted via rubber buffers (type TW70 BL M12, STS Schwingungstechnik Schuster GmbH, Schwäbisch-Gmünd, Germany and type NRE, Netter Vibration, Mainz, Germany) as vibration dampers. The connection cables, optical fibers and coolant lines of all active components in the chamber are led out via standard, closable, side openings in the chamber. They end in a central supply and control unit, whose components are located in a 19' rack-mount housing (RS Pro Series PF19, RS-Components) above the incubator. These central components include two motion controllers (SMC Hydra CM, PI miCos GmbH) for motion control of the XYZ platforms, pump (Laing DDC-1), air-water heat exchanger (Airplex) with fan (Revoltec AirGuard) and control (Poweradjust3) of the liquid cooling system (Aqua Computer GmbH & Co. KG), light source (SST50W, Luminus, Brussles, Belgium) and control unit (SLC-SA Universal LED Controller, Mightex, Toronto, Canada) of the lighting system and their associated power supply components.

All active system components of the array video microscope and its central control and supply units are individually connected to a central control PC via their respective interfaces, the cluster controller via LAN and USB, the controller of the liquid cooling system via USB, and the motion and light controllers each via RS-232 interfaces.

## Operation and control software

On the control PC, a proprietary device control program based on the LabVIEW graphical programming language (National Instruments, Austin, USA) as an integrated operating environment with a graphical user interface ties together all the workflows and higher-level dependencies of the subsystems. The operating procedures are divided into three phases, the basic setup (Session Setup menu), the channel-by-channel image settings (Image Control menu) and the sequence control (Session Control menu). During the basic setup, the well plate exchange, the selection of the active, filled wells and the selection of the well plate type are carried out under software control and guidance. The image settings in the Image Control menu control the stages and the illumination system. Visible well areas can be selected and the image sharpness and illumination of each image channel (well) can be adjusted manually or automatically (auto gain, auto focus). In the sequence control part, the operating mode, timing, storage parameters and monitoring thresholds for video and temperature data recording are set, and recording is started and monitored respectively paused or interrupted. All sequences are error monitored and in the event of an error, the recording is immediately aborted and all moving system components are moved to their respective failsafe positions.

## Imaging speed and data handling

The CMOS image sensor used (OV9712, Omnivision) has a native image resolution of 1280 × 800 pixels (WXGA format) distributed over an

active area of 3888 μm × 2430 μm. At this resolution, a frame rate of 30 FPS is possible, and up to 60 FPS at a reduced resolution (640 × 800 pixels). Per pixel, 30 bits of RGB data are generated. For video generation, the image data is reduced to 720 p format (1280 × 800 pixels) with 8 bits of grayscale. With 16 image sensors this results in ~13.73 GBPS which are down sampled to ~3.3 GBPS by the FPGA. In the process of data reduction, a sharpness indicator is calculated on-the-fly for each frame and only the sharpest image of a time interval of ~1.5 s is stored for each channel, so that ~28.2 MB of RAM is required for intermediate image data storage. If 96-well plates are used, this amount of data accumulates in the frame rate interval of 8 s for the simultaneous generation of 16 videos. If 384-well plates are used, then four times the amount of data is generated in 8 s for the simultaneous generation of 64 videos. During this time interval, the video microscope unit is moved 4.5 mm (center-to-center distance between adjacent wells) four times in the x- or y-direction (travel time <500 ms), then one focusing cycle is performed in the z-direction each time (travel time ~1.5 s), and the image data of 16 wells each time is transferred to the PC. Thereby, two autofocus variants exist. The free running collective autofocus mode and the focus position map-based triggered mode. In the free running collective autofocus mode the drive speed of the Z-stage is independent of the number of recorded channels. In the focus position map-based triggered mode, each new focus position requires a rest time of ~40 ms (Supplementary Fig. 3).

The individual imagers constantly generate images at 30 FPS after system startup, regardless of the selected operating mode of the ComplexEye, with each image being temporarily buffered in the FPGA. This buffering is done according to the "ping-pong" principle, i.e., there are two memories of the same size in the FPGA which are always alternately loaded with the new image data. Thus, data is always written to one of the two memories while the other still contains a complete previous image. In the "focus position map-based triggered" mode the last complete image currently in the FPGA (ping-pong) memory is transferred at trigger time (while at the same time new image data is written to the other of the two memory blocks of the ping-pong memory). In the "free running" mode, a third memory block is needed to store the last sharpest image of an entire focusing sequence. If the sharpness indicator of a new image is greater than that of the last sharpest image of a focusing sequence, this new image is transferred to the third memory block, thus overwriting the previously sharpest image there.

## Cell lines

The cancer cells depicted in Fig. 1b are Cutaneous Melanoma (CM) cells. The CM cells were isolated from the metallothionein-I (MT)/*ret* transgenic mouse model, which spontaneously develops primary malignant melanoma and distant metastasis[47]. The cells were isolated and cultivated by author IH at the clinic for dermatology at the University Hospital Essen, Germany[48]. The long-term recording of migrating cancer cells shown in Supplementary Movie 3 displays CT26 colorectal carcinoma cells. CT26 is a murine colorectal carcinoma cell line which is from a BALB/c mouse. The cell is a clone of the N-nitroso-N-methylurethane-induced undifferentiated CT26 colon carcinoma cell line[49].

## Neutrophil isolation

For the entire experiments, neutrophils were isolated from 1 ml EDTA-supplemented blood using magnetic negative isolation with the MACSxpress® Whole Blood Neutrophil Isolation Kit (Cat. No.: 130-104-434, Miltenyi Biotec, Bergisch Gladbach, Germany) according to the manufacturer's instructions. Residual erythrocytes were also magnetically depleted using the MACSxpress® Erythrocyte Depletion Kit (Cat. No.: 130−098-196, Miltenyi Biotec) according to the manufacturer's instructions. Afterwards, purified neutrophils were washed in sterile PBS, resuspended in sterile hematopoietic growth medium (HPGM) (X-

VIVO™−10 Serum-free Hematopoietic Cell Medium, Lonza, Basel, Switzerland) and automatically counted using a Cellometer Auto T4 (Nexcelom Bioscience, Lawrence, MA, USA).

## Migration assay

For the 16-well experiments at the ComplexEye and for the 4-well experiment at the Leica DMI6000 B, the neutrophil migration assay was performed as previously described[11]. Briefly, purified neutrophils were seeded in a 96 Well μ-Plate (Cat. No.:89621, ibidi, Martinsried, Germany) at a density of 8250 cells per well in 198 μl sterile HPGM supplemented with sterile Serum Replacement 3 (SR3, final concentration: 0.3x, Cat. No.: S2640, Sigma-Aldrich, Munich, Germany). Neutrophils were stimulated with 2 μl fMLP (final concentration: 10 nM; Cat. No.: F3506, Sigma-Aldrich, Munich, Germany), 2 μl human recombinant CXCL1 (final concentration: 100 ng/ml; Cat. No.: 275-GR-010/CF, R&D Systems, Minneapolis, MN, USA), or 2 μl human recombinant CXCL8 (final concentration: 100 ng/ml; Cat. No.: 208-IL-010/CF, R&D Systems). As all stimuli were reconstituted in sterile PBS, the addition of 2 μl PBS alone served as a vehicle control. The plates were centrifuged (50 g, 3 min) and incubated at 37 °C for 20 min before microscopy. For the 64-well experiment at the ComplexEye neutrophils were seeded in a 384-well μ-plate (Cat. No.: Z722995, Thermo Scientific, Dreieich, Germany) at a density of 2000 cells per well in 48 μl sterile HPGM supplemented with sterile SR3. The cells were stimulated in the same way as described before.

## Time-lapse microscopy

Almost all samples were imaged in the ComplexEye with a 4.7X magnification and rate of one frame/8 s for 1 h at 37 °C. Only one sample was recorded with the conventional Leica DMI6000 B (Leica Microsystems, Wetzlar, Germany) with a motorized stage, 13.4X magnification, and rate of one frame/8 s for 1 h at 37 °C to compare the performances.

## Compound screening

1000 compounds from a library with known bioactives were provided by the Lead Discovery Center (LDC; Dortmund, Germany). The compounds belong to a very heterogenous group of substances and are by and large in clinical development or launched. For the screen purified neutrophils were plated in a 384-well μ-plate at 2000 cells/well in HPGM + SR3, followed by the addition of 28 μl compound (final concentration: 5 μM) in DMSO (final concentration: 0,1%) and 2 μl fMLP (final concentration: 10 nM). The ComplexEye allowed us to record 64 wells in one run with 3 controls in each round. To have a baseline of the fMLP stimulated neutrophil migration in each round, 10 nM fMLP alone (provided as 2 μl of a 150 nM stock solution) was the first control. As all compounds were reconstituted in DMSO, the addition of 2 μl DMSO alone in the same final concentration served as second control in half of the screen. In the other half fMLP + DMSO was the second control. The last control was the addition of 2 μl PBS only to have a baseline for unstimulated neutrophil migration. With three controls in each round we were able to screen 61 wells with different compounds in one run with the rate of one frame/8 s for 1 h. With this the screening of 1000 compounds was completed in 17 rounds and 4 days with only 8 donors. Thereby, hit compounds were identified based on a threshold. The compound where the cells migrated at least 40% slower than the fMLP control in the same round was defined as a hit. In the first round, all compounds were measured in singlets. After identifying multiple hit compounds, they were validated in two additional rounds.

## Autotracking

The generated movies were exported as *.avi files. With these files automated segmentation was performed. Segmented masks were obtained by applying a fine-tuned model of the cellular segmentation tool Cellpose[50] to the recorded movies. These masks were then

analyzed using a tracking algorithm based on Earth Mover's Distance[51]. To evaluate the results, 14 pairs of consecutive frames from various movies were randomly selected and inspected. The occurring errors of the tracks were counted manually. This was done on the basis of the segmented mask of the former frame (i.e., errors in the segmentation of the former frame were not considered in the tracking evaluation). A total of 2207 tracks were created by the algorithm, of which 7 were erroneous. 43 of the cells correctly detected in the first frames were not tracked, 20 cases of which were due to false negative segmented cells in the later frame. In short, 97.78% of the tracks in the evaluated frames were correct. In terms of segmentation, 97.39 % of the cells were detected correctly in these frames

## Statistics and reproducibility

Sample sizes were always chosen based on the individual experiments. No data were excluded from the analyses. For the performance comparison of the ComplexEye with a conventional microscope (Fig. 3a) the sample size was chosen to be $n = 1$. Here, we wanted to compare the performance of both microscopes by simultaneously measuring the cells from the same donor. We then also compared the results with previously published values ($n = 25$). In Fig. 3b, we chose $n = 1$ with 16 technical replicates for each condition (PBS, fMLP, CXCL1, CXCL8) as the aim of this experiment was to demonstrate the homogeneity of the values obtained from a single donor. Next, in Fig. 3c we wanted to demonstrate that the ComplexEye is able to measure cells from 16 donors, each under 4 conditions. Therefore, an $n = 16$ was used here. The initial screen of the 1000 compounds was performed with an $n = 1$ to analyze the compounds in HT. After identifying 17 hit compounds, these compounds were validated with an additional $n = 2$. In our experiments, the hit rate in the primary screen was 1.7% (17 out of 1000 compounds). The validation rate of these hit compounds was 70.6% (12 out of 17 compounds). Results in all experiments that were not repeated were highly consistent with the results from other experiments in this study and also from a previously published study. As the tracking results were generated by an automated software, the investigators were not blinded during data collection. Therefore, investigator bias is not considered to contribute to the data. All statistical analyses and plotting were performed using GraphPad Prism™ (Version 9.0, GraphPad Software, San Diego, CA, USA). Experimental data were plotted as bar graphs. Statistical computation, such as computation of $p$-values was performed using the Kruskal–Wallis test with Dunn´s multiple comparisons test. All data are presented as median values ± interquartile range. *$p < 0.05$, **$p < 0.01$, ***$p < 0.001$ unless noted otherwise. The statistical tests are also described in the respective figure legends.

## Reporting summary

Further information on research design is available in the Nature Portfolio Reporting Summary linked to this article.

## Data availability

To reproduce our results, raw data of all movies tracked in the frame of this manuscript have been deposited at https://doi.org/10.5281/zenodo.7962144. Source data are provided with this paper. Some images in Figs. 1, 2 and 4 were created with Biorender.com. Source data are provided with this paper.

## Code availability

The code used for tracking of cells as described in methods can be found at https://github.com/MMV-Lab/complex_eye_analysis[52].

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

## Acknowledgements

This work was supported by a grant from the Mercator foundation (Anschubförderung An-2014-0050) as well as intramural funds from the University Duisburg Essen (UDE) and the medical faculty of the UDE to M.G., A.G. and R.V. Further support of the work came from the German Research Foundation (DFG) to M.G. (CRC TRR332 TP C6, KFO 337 "PhenoTime" TP7 (also to I.H.), FOR-2879 "Immunostroke" project 405358801 and GU769/10-1). Further funding was provided by the Bundesministerium für Bildung und Forschung (BMBF) and by the Ministerium für Kultur und Wissenschaft des Landes Nordrhein-Westfalen (MKW). We thank Jürgen Becker and Kathrin Blank for blood drawing and support during this study. We acknowledge support by the Open Access Publication Fund of the University of Duisburg-Essen. All members of the laboratories are acknowledged for intense discussions.

## Author contributions

M.G. conceived the idea and assisted R.V. and A.G. in realization of the working model. R.V. conceptualized the cluster controller with CMOS chips under the lenses and FPGA outside the lens zone and realized the system and the control software together with A.A.T., V.S., J.H., S.O., G.H., M.R., P.R., K.Sm., R.B. and K.Se. Z.C. performed all biological experiments with the help of A.B., A.K., I.H., S.G. and A.S. A.-K.K., B.K., J.E.E. provided the compounds, essential knowhow for the screen, as well as analysis and clustering of the initial screening hits. J.S., L.K. and J.C. developed the tracking algorithms and analysis of the resulting data. M.G. wrote the paper together with R.V., Z.C. and J.H. All authors contributed to proofreading. We thank the workshops at the University Duisburg Essen as well as the ISAS for producing essential building blocks of the system. The Imaging Center Essen is acknowledged for expert support. Martin Collischon (Collischon Optik-design, Erlangen, Germany) is acknowledged for delivering the illumination and focusing system and help in debugging the internal lens optics.

## Funding

## Competing interests

M.G., R.V., S.O. and A.G. have filed a patent application in Germany and worldwide (WO2018019406A3) for the imaging concept of multi-lens video microscopy as described in this work. All other authors declare no competing interests. A.-K.K., J.E.E. and B.K. are employees of the LDC GmbH and have no competing interests.
