## [Peer Review File · Nature Communications]

Reviewers' comments:

Reviewer #1 (Remarks to the Author)

The paper, titled "ComplexEye - a multi lens array microscope for High-Throughput embedded immune cell migration analysis," introduces a new array microscope called ComplexEye that is designed to efficiently analyze immune cell migration in high-throughput applications. While the system does offer some advantages, it is unclear whether it is significantly better than conventional microscopy for studying cell migration. One major issue is that the system lacks fluorescence microscopy capability, which is essential for many applications. Additionally, the energy efficiency of the system may not be a significant factor in a laboratory setting. Previous cell migration studies have already demonstrated a comparable scale using conventional microscopy to track individual cells, and the ComplexEye system cannot study chemo-taxis, an important aspect of cell migration. Therefore, it is unlikely that this technology will have a significant impact on microscopy or cell migration studies. Furthermore, the presented cell migration study does not provide any immediate advancement in clinical translation. Overall, the paper fails to convincingly demonstrate that the ComplexEye system will revolutionize microscopy or significantly advance our understanding of cell migration.

1. As discussed by the authors, "ComplexEye only enables the acquisition of bright-field images." Lack of fluorescence microscopy is a huge disadvantage as compared to conventional microscopy.

2. One important aspect of cell migration that cannot be studied using the proposed method is chemo-taxis.

3. The field of view offered by ComplexEye appears to be limited. To improve the paper, the authors should provide a quantitative comparison of this aspect. For instance, they could compare the field of view with that of the ECLIPSE Ti2 inverted microscope, which provides a 25mm field of view.

4. Although the authors claim that their method is energy-efficient, energy consumption is not a crucial factor for a laboratory instrument. Taking into account the three previous comments, the overall evidence is unconvincing that this method is substantially superior to conventional microscopy.

5. Recent studies have shown experiments on cell migration at a comparable scale, where conventional microscopy was used to track individual cells. The presented technology does not seem to have a significant impact on the investigation of cell migration.

"High Throughput Confined Migration Microfluidic Device for Drug Screening"

"High-Throughput Cellular Heterogeneity Analysis in Cell Migration at the Single-Cell Level"

6. The authors fail to provide novel insights into biology with the screening experiment demonstrated in this work. However, if they can prove the significance of something new and validate it through animal or clinical results, their paper would be considerably strengthened.

7. It is suggested that the authors provide the correlation coefficient between the motility of identical conditions on two plates. Additionally, the authors should quantify the likelihood of false discovery. This would involve determining the probability that a treatment which enhances cell migration is not detected, as well as the probability that a treatment with no effect is mistakenly identified as a migration booster or inhibitor. These quantitative values would enhance the reliability of the migration screening results and improve the audience's comprehension of the experiment.

Reviewer #2 (Remarks to the Author)

In this manuscript, the authors describe a composite microscope consisting of an array of detectors, each aligned with its own detection lens and illumination system. The microscope is capable of imaging 16 regions of interests in parallel and is designed to specifically image well-plates. The microscope relies on custom optics, housing, electronics and software. All designs (optics, electronics and microscope body) and control code (microscope control and data acquisition) seem to be closed-source. The authors use their instrument to perform screening of bioactive compounds impact on the migration of neutrophils at a higher throughput than possible with single-lens instruments. Finally, the manuscript is quite clear and well written.

The microscope appears to be a fit of engineering, combining smart ideas (moving frame rather than sample to not disturb the latter, insulating the electronics from the sample environment, distributing the light with polymer waveguides, computing focus metrics on-the-fly) and complex devices (embedded FPGA with multiple detectors, custom detection lenses). It successfully overcomes the shortcomings of traditional commercial microscopes for this particular application (imaging speed, stability, throughput). While I am not equipped to judge the clinical impact, the authors convincingly demonstrate that their microscope can turn tedious and lengthy studies into more feasible tasks.

Overall, I would have hoped to find more quantitative measurements of the instrument capacities and limitations (e.g. temporal and spatial resolution). Additionally, the manuscript would have a much higher impact would the design files be open to the community, thus allowing for reproducibility of the research. Yet, I understand that the instrument has some commercial potential and that a patent application has been submitted.

In my opinion, the manuscript is conform to Nature Communications publication criteria and should be accepted with minor changes. I have organized my comments between major (important comments I would like to see addressed) and minor comments (small remarks).

Major comments

1. The authors state in the results that “The resolution was numerically ~74% that of the standard system (2.1 px/μm vs. 1.55 px/μm in ComplexEye).” In this case this is the pixel resolution (akin to magnification) and should not be mistaken for optical resolution. I encourage the authors to make this point clear to avoid misunderstanding. The authors should also compute the magnification of their custom optics from the pixel resolution and indicate it, in order to help readers better grasp how it compares with the commercial instrument and its 20x lens. In addition to magnification (as they are related), the authors should also mention the total field of view size (pixel resolution x sensor size).

2. Figure 1c is cited as showing that the microscope leads to image that are “optically similar” to that of a commercial instrument. The images in 1c only show that the two systems have different magnifications, and even that fact is not self-evident as the two images are scaled to similar size and only the number of pixels gives it away.

The authors should use a USAF pattern (probably a negative one for bright field, see Resolution Test Targets in Thorlabs for examples) as this would constitute a much better comparison of optical resolution/quality.

In addition, using a USAF pattern would allow them to quantify the contrast in different parts of the field of view, which could be a supplementary figure.

3. The temporal resolution of the experiments is a key consideration in this study, as the authors insist on the 1 image / 8 seconds (per well) limit under which neutrophils can effectively be tracked. Without clear breakdown of the different steps of the imaging in terms of time (stage travel speed, exposure time, refocusing), it is difficult to follow the comparisons with previous work and to easily grasp the temporal limitations.

For instance, the authors write in the introduction: "The staggered lenses required 8 steps to image all wells of a 96-well plate[31], still double the time required for immune cell imaging in 96-well plates and 8-times too slow for 384-well plates.". This statement doesn't make sense without telling the reader how long each of the steps took for imaging. If each of the 8 steps took one second only, it could be sufficient to image a 96 well plate at the desired temporal resolution.

The other aspect of this statement that I find troubling, is that it seems to imply that the other published instrument is capable of imaging 48 wells of a 96 well plate at the required speed (since it images the whole plate 2x too slow, it can image half of it at the required speed), well over the authors' system (16 for a 96 well plate) if we are to believe the following sentence a paragraph later: "ComplexEye [...] to simultaneously image 16 adjacent wells of a 96 well plate or 64 wells of a 384-well plate with one frame per 8 seconds". Likewise, the imaging of a 384 well plate by the other published microscope is allegedly 8 times too slow, meaning that it could potentially also image 48 wells of a 384 well plate within the required speed. However, the well to well distance in a 96 well-plate is twice that of a 384 well-plate. Therefore if stage travel time is a major factor limiting the imaging speed, they should be able to image more regions in the 384 well-plate. Without additional explanation, the statement could be considered misleading.

Finally, I am puzzled why ComplexEye seems to be limited to imaging only 16 of the 96 wells on a 96 well plate at the necessary imaging speed? Is it an actual limitation or simply the maximum imaging speed used by the authors in the present manuscript?

All in all, the authors should clarify these points and discuss in more details the typical exposure time used with such samples, the time needed for refocusing, the potential need for a rest period after moving the stage (if applicable) and the typical stage travel speed considered "safe" for imaging neutrophils. This would help clear potential confusion and put in perspective the numbers (1 frame / 8 seconds, number of wells that can be imaged). An additional figure illustrating the factor limiting spatio-temporal resolutions (typically stage travel speed, refocusing) could be useful and improve the reading experience.

4. The authors state that the use of white LED allows "generating a homogeneous illumination (Fig. 1d)". But Figure 1b clearly has a background gradient. The authors should provide a supplementary figure showing a side-view intensity plot of the illumination and justify the "homogeneity" statement quantitatively. Is this pattern similar in all 16 illuminations?

5. The following sentence is complex and not entirely clear: "the whole 16-fold lens unit was moved periodically up and down within the overall focus range while in every round each of the 16 imagers took a picture in the moment of maximum image sharpness at the predefined frame to frame time-gap of the resulting movie".

The most logical deduction to me is that for each well, the authors calibrated before imaging an optimum focus for the corresponding lens. For each position of the lenses array, the distribution of focus positions among the lenses defines a focus range. Then, during imaging, at each position, the stage

goes through that range and the sensors only take pictures at the calibrated focus position of their corresponding lens.

This interpretation is not compatible with the “Data handling” part of the methods, which seems to imply that images are recorded at every focus position and only the maximum sharpness image is kept in the FPGA.

The authors should make their statement clearer. A supplementary figure illustrating the imaging protocol would be very useful. Finally, in order to satisfy the more optics-minded readers, the authors should provide a supplementary figure showing an example of focus distribution between lenses (e.g. stage height distribution while imaging the same well with each lens) and across wells (e.g. imaging all wells with a single lens and plotting the stage z position distribution).

6. I did not find the “Data availability statement” in the manuscript. The only dataset provided is a very small example used to illustrate the pipeline. Given the argued impact of the high-throughput study, the authors should provide the raw images and scripts to allow reproducing their results, for instance by depositing them on Zenodo.

Minor comments

1. What is the power of the illumination used on each illuminated area, or better the illumination power per square centimeters?

2. Figure 1b is missing scale bars.

3. As the authors state: “To protect the electronics from the humidity in the incubator, we encapsulated them to separate them from the sample level (Fig. 2c)”. The authors could specify by which means does the encapsulation protect the electronics from humidity (sealing?). How is dryness maintained in the box, simply by insulation?

4. “The analysis showed that the measured values were comparable to our migration data obtained before”. Could the author explicitly state the values obtained in their previous work? And potentially make them apparent in figure 3a.

5. On Fig 3, the authors should mention the number of tracks per points on average (and potentially with a standard deviation).

6. Figure panels 4a and 4b are too small, I would advise the authors to split the figure into two to allow more comfortable reading. Consequently, this would lead to less crowded figures.

7. The authors should provide a rough estimate of the price of their custom instrument.

Responses to reviewers for “ComplexEye - a multi lens array microscope for High-Throughput embedded immune cell migration analysis” by Cibir et al.

Response to Reviewer 1.....Page 2
Response to Reviewer 2.....Page 13

We would like to thank the reviewers and the editor for the positive evaluation of our manuscript and the constructive comments. Based on these comments, we have performed many additional experiments, included new data and have revised the manuscript accordingly. We feel that with the help of the reviewers’ comments the revised manuscript has been significantly improved and hope that it will now be acceptable for publication. Please find below a point-by-point reply to all individual comments.

Reviewer 1:

The paper, titled "ComplexEye - a multi lens array microscope for High-Throughput embedded immune cell migration analysis," introduces a new array microscope called ComplexEye that is designed to efficiently analyze immune cell migration in high-throughput applications. While the system does offer some advantages, it is unclear whether it is significantly better than conventional microscopy for studying cell migration. One major issue is that the system lacks fluorescence microscopy capability, which is essential for many applications. Additionally, the energy efficiency of the system may not be a significant factor in a laboratory setting. Previous cell migration studies have already demonstrated a comparable scale using conventional microscopy to track individual cells, and the ComplexEye system cannot study chemotaxis, an important aspect of cell migration. Therefore, it is unlikely that this technology will have a significant impact on microscopy or cell migration studies. Furthermore, the presented cell migration study does not provide any immediate advancement in clinical translation. Overall, the paper fails to convincingly demonstrate that the ComplexEye system will revolutionize microscopy or significantly advance our understanding of cell migration.

1. As discussed by the authors, "ComplexEye only enables the acquisition of brightfield images." Lack of fluorescence microscopy is a huge disadvantage as compared to conventional microscopy

We agree that it would be advantageous to incorporate fluorescence analysis in the workflow of ComplexEye. However, this is technically very challenging to solve. In a classical single lens microscope, the light path has an angle of 90° between illumination and detection. This allows easy integration of a dichroic mirror/beam splitter that reflects the very bright illumination light from a LED or high-pressure mercury lamp into the sample while filtering it out efficiently from the red-shifted fluorescent light emitted from the sample before this enters detectors or a camera. The multi objective setup of the ComplexEye does not allow such a construction due to spatial restrictions under the individual lenses. Hence, the only practicable way to incorporate fluorescence in a multi-lens array is the use of a direct light path without an angle. Illumination in this setup is trivial, e.g. by using individual coloured LEDs whose light is fed into the system by the same light guides that we currently employ for white light illumination. However, filtering can only be achieved by bandpass filters in front of the lenses or the imaging sensors. These filters must be able to separate extremely bright illumination light from the red-shifted fluorescence in the sample which is much dimmer (by a factor of 10^4 up to 10^{11} , as described in the handbook of optical filters (<https://www.chroma.com/sites/default/files/HandbookofOpticalFilters.pdf>)). Other than in a 90° angled setup, where the filtering capacity of the beam splitter is combined with another filter in front of the detector, in a direct path setup only this one filter is available. Hence, it

requires a huge effort to find a suitable combination of illumination and filtering in ComplexEye. While we acknowledge, that this would be very helpful, it is still beyond the scope of the current study to incorporate it. It requires a completely new construction to be realized. We still are confident, that the performance of our high-throughput bright field microscope has a large number of useful applications that work without fluorescence. Tracking the migration of human neutrophils, as demonstrated by us, is just one of them. Any other immune cell or other cell type can also be studied with it with unprecedented throughput and also chemotaxis, as we show below (see answer to point 2).

2. One important aspect of cell migration that cannot be studied using the proposed method is chemotaxis.

We thank the reviewer for pointing out this important aspect of cell migration. Although we have not particularly focussed on chemotaxis in our analysis, it is also possible to analyse this with ComplexEye. The only principal difference between a chemokinesis experiment (our data) and a chemotaxis analysis is that in the latter a migration trigger is offered as a gradient that migrating cells can sense, while in the former the migration trigger is present everywhere at the same concentration. The general microscopic imaging for both scenarios can be identical. To demonstrate that measuring chemotaxis with our ComplexEye microscope is principally possible, we have developed a simple chemotaxis assay using heparin beads as a source of fMLP. For this, heparin beads from Adar Biotech (Cat. 6024-5) were incubated with 100 μ M fMLP in PBS for 2 hours. Control beads were incubated with just 1x PBS. Afterwards, both types of beads were washed twice in 1x PBS. Next, neutrophils from a healthy volunteer were isolated and plated in 96-well plates. 5 μ l from the bead solution containing approximately 20-50 beads with either fMLP or PBS were added to the neutrophils. The cells were recorded with ComplexEye for 1 hour with 8 seconds between frames. From this chemotaxis experiment, one new Supplementary Movie 6 and two new Supplementary Figs. 4 and 5 were generated (reproduced for the reviewer below). The Supplementary Movie and the tracking results clearly demonstrate, that the neutrophils ignore PBS beads while they are attracted to the fMLP beads to which they migrate with much higher values for directed speed and directness compared to control neutrophils. Thereby, the directed speed is defined as a change in Euclidean distance (a vector between a cell track's end position and its start position) divided by the time unit. Therefore, the directed speed is an indicator of a chemotactic effect. This novel chemotaxis assay demonstrates, that analysing chemotaxis is principally possible with our microscope. After optimization, the beads could be incubated with any chemokine or cytokine, allowing chemotaxis assays to be performed with any type of cell.

Supplementary Fig. 4 | **Chemotaxis assay with beads.** Freshly isolated neutrophils from healthy donors were plated on a 96-well plate and heparin beads (Adar Biotech (6024-10) incubated either with 100µM fMLP or 1x PBS (control) for 2 hours and washed twice with 1x PBS were added to the neutrophils. Neutrophils plus beads were recorded for 1 hour with 8 seconds between frames. The tracking results clearly demonstrate that neutrophils are attracted to the fMLP beads whereas PBS beads did not recruit the cells. Bars are given as median \pm interquartile range, $n=2$.

Also, thanks to the helpful notion of the reviewer, we were able to make a new interesting discovery during the chemotaxis assay. Neutrophils co-cultured with fMLP beads displayed a normal polarized migration morphology until they got into contact with the fMLP-beads. Shortly (~2 min) after the cells touched the fMLP-beads, they completely changed their morphology from a normal roundish shape to an extremely spread-out flattened shape within about 2 minutes. This flattened morphology could retract back to normal morphology before cells left the beads again. We demonstrated this morphology change in the new Supplementary Fig. 5 (reproduced for the reviewer below). We have also updated the MS text. **The text now reads:** "Furthermore, besides measuring chemokinetic migration, ComplexEye is also able to investigate chemotaxis. In a bead-based assay we also discovered a very unusual behavior of human neutrophils when encountering fMLP from a solid source as opposed to its availability only as a soluble factor (Supplementary Figs. 4 and 5 and Supplementary Movie 6)."

Supplementary Fig. 5 | **Neutrophils change their morphology when they get in contact with fMLP beads.** Freshly isolated neutrophils from a healthy donor were plated on a 96-well plate and heparin beads incubated with 100µM fMLP for 2 hours and washed twice with 1x PBS were added to the neutrophils. **a**, Exemplary images of neutrophils co-cultured with fMLP beads. One individual neutrophil (red circle) was followed over time. The neutrophil displays

*normal amoeboid shape until it gets in contact with the bead. Upon contact, it changes the morphology and demonstrates a flattened and elongated shape. Afterwards it turns over to an amoeboid shape again and leaves the bead. Time shown in hh:mm:ss. **b**, The graph displays the behaviour of 4 individual neutrophils over time. The red line values belong to the neutrophil from the exemplary images in a. **c**, The table shows the duration of each phase for the individual neutrophils in minutes.*

Hence, this assay shows that high concentrations of fMLP from a solid source induced drastic but transient cell changes that led to huge membrane proportions of neutrophils being in touch with the fMLP source, while soluble fMLP induced the typical amoeboid shape of freely migrating cells. The assay allowed to precisely measure the transition between both states and the related timing between the different shapes. It was not known before that human neutrophils are able to differentiate between the physical state of the same migration trigger and respond to this with massive morphological adjustments within minutes.

3. The field of view offered by ComplexEye appears to be limited. To improve the paper, the authors should provide a quantitative comparison of this aspect. For instance, they could compare the field of view with that of the ECLIPSE Ti2 inverted microscope, which provides a 25mm field of view.

We thank the reviewer for this notion. Indeed, Nikon advertises a field of view (FOV) of 25 mm for the ECLIPSE Ti2. However, it must be kept in mind that this FOV is only valid for a magnification of 1x (we have communicated with Nikon about this, and they have confirmed it). However, Nikon does not sell a 1x lens for the system. The smallest lens available for the system is a 2x, which reduces the FOV to 12.5mm (the effective FOV is the field number (FN) divided by the magnification of the lens. Hence $25\text{mm}/2x=12.5\text{mm}$). At 2x magnification it would be possible to see more than an entire well of a 96-well plate. However, individual cells would be so small (the optical resolution of the lens with a NA of 0.1 is $3.35\ \mu\text{m}$, thus mapping one neutrophil to just 2 pixels of an image), that no details would be visible and tracking would become very challenging (the movement even of the fastest cells would be 6-7 pixels per minute. Essential small movements would remain undetected). With a 20x lens, which compares best to the ComplexEye lenses, the ECLIPSE Ti2 would have a FOV of 1.25 mm. The ComplexEye microscope has a FOV of $825.8\ \mu\text{m}$ (see also answer to R#2 below for a precise description of how we measure this value). Hence, ComplexEye does indeed have a 34% smaller FOV compared to the commercial ECLIPSE system, but this is due to the technical limits of the 96-well plate. As of today, the size of available CCD/CMOS chips that are small enough to fit under single wells of 96 well plates do not allow to incorporate larger FOVs. In the future chips with larger light-sensitive areas might allow larger FOVs to be

realized, but at the time of construction 825.8 μm was the maximum doable size. It is important to note, that for tracking the migration of neutrophils 825.8 μm FOV are by far large enough, as we see on average 150 individual trackable neutrophils per frame in a Movie, hence delivering very robust migration data from each Movie. Seeing 34% more cells would not generate significantly better migration data. But an ECLIPSE Ti2 system alone, even though it has a greater FOV and even when combined with an electric stage, can only deliver 4 independent Movies of migrating neutrophils in different wells per time (with the necessary time resolution of one frame per 8 seconds), while ComplexEye delivers 16-fold more. For screening large chemical libraries this is a much more relevant feature than a large FOV.

4. Although the authors claim that their method is energy-efficient, energy consumption is not a crucial factor for a laboratory instrument. Taking into account the three previous comments, the overall evidence is unconvincing that this method is substantially superior to conventional microscopy.

With all due respect, we do not agree with this view. In times of rising CO₂ emissions, any approach that can help reduce the carbon footprint of a method is welcome and necessary. If it is possible to get the same amount of information from a scientific experiment while using over 30 times less energy, we consider this to be extremely important. We would also like to bring this kind of thinking to the community. As scientists we all are constantly improving our methods and approaches. Looking at the energy consumption of devices has not been an issue so far, but we strongly believe that it should be, and ComplexEye proves that by optimizing energy consumption, it is even possible to significantly improve the throughput of a method.

5. Recent studies have shown experiments on cell migration at a comparable scale, where conventional microscopy was used to track individual cells. The presented technology does not seem to have a significant impact on the investigation of cell migration.

"High Throughput Confined Migration Microfluidic Device for Drug Screening"

"High-Throughput Cellular Heterogeneity Analysis in Cell Migration at the Single-Cell Level"

In the publication "High Throughput Confined Migration Microfluidic Device for Drug Screening"¹ the authors introduce a microfluidic device for analysing the migration of different cancer cell lines. Thereby, they show that this device reduces reagent consumption and enhances throughput. While this is indeed true for cancer cell migration analysis, the device would fail for cells that migrate significantly faster than cancer cells such as the neutrophils investigated by us. In their study, Yang et al. performed real-time tracking of cells for 24 h with 30 minutes between frames. For cancer cells, recording one frame per 30 minutes is sufficient to capture their movement. Immune cells are ~20-100 times faster, especially when they are

stimulated with chemokines. In particular, neutrophils that are stimulated with bacterial products such as fMLP migrate at 15-20 $\mu\text{m}/\text{min}$ (among the fastest migrating cells in the human body), thus allowing not more than eight seconds between frames to enable effective tracking. Therefore, conventional microscopes as used in the above-mentioned publication, can only image four wells at the speed required for neutrophil tracking. In contrast, the ComplexEye system is able to image 64 wells under the same conditions. Furthermore, Yang et al. only analysed the migration area in μm^2 and did not quantify important parameters such as speed and amount of moving cells. Besides this, they investigated cell clusters whereas we are analysing motility at the single cell level. With this we are able to additionally detect cell morphology changes, which delivers important information on cellular behaviour as shown above for chemotaxis. With just one frame per 30 minutes the entire dramatic morphological change of neutrophils in response to bead-bound fMLP would have gone unnoticed, as it occurs within less than 30 min. Taken together, Yang et al successfully established a high-throughput microfluidic platform to analyse cancer cell migration which is indeed very interesting and important but their device would not be suited to investigate immune cell migration in high throughput like ComplexEye can.

In the other publication “High-Throughput Cellular Heterogeneity Analysis in Cell Migration at the Single-Cell Level”² the authors introduced a different and very interesting device to analyse the migration of individual cancer cells in microchannels. Thereby, the motility of single cancer cells was investigated in two ways. First, the migration distance was measured based on the final cell frontier (the cell migrating the farthest) of each migration channel after 24 h of incubation. This means the authors did not perform real-time recording of the migrating cells but instead took an image at 24 hours and defined the cells which had migrated the farthest. In another assay they recorded the migration of cells by imaging them once per hour for 13 hours. The measured values of the fastest cells in their assay was 390 μm in 13 hours, which equals to 0.5 $\mu\text{m}/\text{min}$. The mean value was $\sim 50\%$ of that. Hence, even the fastest cells in these assays are 30-40 times slower and their mean migration in a group is 60-80 times slower than fMLP-stimulated neutrophils. We appreciate that this study offers interesting new possibilities for analysing the migration of individual cancer cells but we must also emphasize that the system used by the authors is not comparable to our device. The setup introduced in this paper would not be suitable to track single migrating immune cells.

6. The authors fail to provide novel insights into biology with the screening experiment demonstrated in this work. However, if they can prove the significance of something new and validate it through animal or clinical results, their paper would be considerably strengthened

We do not fully agree with this view. We do provide new insights into biology with our screening since we were able to identify and now also validate (see next point) several compounds that

have a strong effect on neutrophil motility. Most compounds from the screen were either in phase 2/3 trials or already launched. They are employed for various diseases such as cancer, neurological diseases and infections. After searching for the 5 validated compounds (see next section) we were not able to find any information about their effect on neutrophil migration. It is not known, that these compounds strongly affect the normal motility of neutrophils. The ability to migrate is very important for neutrophil functions. If they are prevented from migrating, they cannot fulfil their function as first line of cellular defence. This could also lead to neutropenia and thus to immunosuppression which might have serious side effects. Therefore, we provide essential novel insights into drug action. Validating these results in an animal experiment, despite certainly being valuable, would, however, require developing a new animal model and seeking for approval to perform this by the authorities, which is impossible to be completed in a reasonable time frame. Hence, we consider this beyond the scope of the present work.

7. It is suggested that the authors provide the correlation coefficient between the motility of identical conditions on two plates. Additionally, the authors should quantify the likelihood of false discovery. This would involve determining the probability that a treatment which enhances cell migration is not detected, as well as the probability that a treatment with no effect is mistakenly identified as a migration booster or inhibitor. These quantitative values would enhance the reliability of the migration screening results and improve the audience's comprehension of the experiment.

This is a very important suggestion. The correlation coefficient between the motility of the same neutrophils under identical conditions on two plates can only be achieved by simultaneous measurements on two independent microscopes. This experiment was already performed and the results are shown in figure 3a. Here, neutrophils from the same donor were treated exactly in the same way and recorded simultaneously at the ComplexEye and a conventional Leica microscope. We have now measured the correlation coefficient for this experiment and find it to be $r=0.9928$ for the amount of moving cells and $r=0.9995$ for the speed (a). Since for this experiment we had to use two independent microscopes, we now also performed another experiment, that only uses ComplexEye. For this we investigated the same cells on two different plates, yet consecutively. This means that the cells of the second round had to be incubated on the plates for one hour before they could be measured. The tracking of these cells led to a correlation coefficient of $r=0.9368$ for the amount of moving cells and $r=0.9480$ for the speed (b). It is important to note, that neutrophils do age in culture and hence a one hour incubation time is very likely to introduce small changes in the motile response, as can be seen in the experiment. However, taken together the data show very high correlation of independent measurements and hence demonstrate the robustness of the assay and that it is

largely independent from external factors such as the individual 96-well plate (as long as plates from the same supplier and catalogue Nr. are used throughout experimental series).

Figure for Reviewer only:

Motility of the same neutrophils on different plates

a, to identify the correlation coefficient between the motility of the same cells under identical conditions on two plates, neutrophils from a single donor were isolated. The cells were prepared exactly in the same way and recorded simultaneously at the ComplexEye (CE) and a conventional Leica microscope. **B**, to examine the correlation of the motility of the same neutrophils that were recorded consecutively at the ComplexEye, neutrophils from another donor were isolated. Here, one plate was recorded at the ComplexEye whereas the other plate was incubated for the duration of the recording (1 hour) in the incubator. Importantly, the stimuli were added just before the second plate was placed under the microscope.

In addition, we wish to emphasize that our system can also identify compounds that increase the speed of neutrophils. In our screen we have identified 9 such substances that increased

the speed by more than 20%. The relevant compounds and their migration enhancing capacity are now listed in the new Supplementary Table 2 and reproduced for the reviewer below.

Cpd name	Chemical target	% speed increase
R6 E5	Reverse Transcriptase	26.6
R6 E3	GSK-3	24.9
R6 D4	Thyroid Hormone Receptor	23.1
R6 E7	Apoptosis	22.5
R6 D2	Apoptosis; c-Met/HGFR;	20.8
R10 A2	Apoptosis; Histone Methyltransferase	20.5
R10 E1	Apoptosis, mitochondrium	20.5
R7 F1	MMP	20.4
R7 A3	Antibiotic; Apoptosis; Autophagy	20.0

Supplementary Table 2. **Overview of the chemical targets of speed-increasing compounds**

Abbreviations:

GSK-3: Glycogen synthase kinase 3
cMet/HGFR: Tyrosine-protein kinase Met/hepatocyte growth factor receptor
MMP: Matrix metalloproteinase

Furthermore, to determine the probability that a compound with migration inhibitory function is mistakenly identified in our screen, we repeated key experiments with these compounds. The amount of substance available from the screened compound library only allowed single Movies and no additional validation step. From our list of hits in this screen we could purchase 8 out of 17 migration inhibiting compounds in greater quantities for further validation, as requested by the reviewer. These 8 compounds were now tested in three rounds with freshly isolated neutrophils from two healthy volunteers. Here, the migration decreasing capacity of 5 compounds could be validated whereas with 3 compounds we could not reproduce the initial effect (see figure below). The 5 compounds were indeed confirmed to decrease the fMLP stimulated speed by more than 60%. This highlights the importance of validating the identified compounds in additional experiments and we wish to thank the reviewer for this important request.

Figure for Reviewer only:

Validation of compounds

To determine, whether the speed-decreasing compounds identified in the first round of the screening would show the same effect in additional experiments with neutrophils from different donors, the compounds were tested in another two rounds with neutrophils from two different healthy volunteers. Thereby, 5 out of 8 tested compounds were confirmed in their speed decreasing behaviour and treated neutrophils from different donors also displayed strongly reduced speed relative to the fMLP control. However, for 3 of the compounds we could not demonstrate a speed decreasing effect. This could point to neutrophil-specific responses that affect only a subpopulation of individuals, while the 5 validated compounds are likely to be broadly effective on neutrophils from any donor. Black circles show the new data of n=2 healthy volunteers and green points display the data of the same compound as found in the initial screen. The red dashed line indicates 40% speed reduction.

References for Reviewer #1:

- 1 Yang, Z. *et al.* High Throughput Confined Migration Microfluidic Device for Drug Screening. *Small* 19, e2207194, doi:10.1002/smll.202207194 (2023).
- 2 Zhou, M. *et al.* High-Throughput Cellular Heterogeneity Analysis in Cell Migration at the Single-Cell Level. *Small* 19, e2206754, doi:10.1002/smll.202206754 (2023).

Reviewer 2:

In this manuscript, the authors describe a composite microscope consisting of an array of detectors, each aligned with its own detection lens and illumination system. The microscope is capable of imaging 16 regions of interests in parallel and is designed to specifically image well-plates. The microscope relies on custom optics, housing, electronics and software. All designs (optics, electronics and microscope body) and control code (microscope control and data acquisition) seem to be closed-source. The authors use their instrument to perform screening of bioactive compounds impact on the migration of neutrophils at a higher throughput than possible with single-lens instruments. Finally, the manuscript is quite clear and well written.

The microscope appears to be a fit of engineering, combining smart ideas (moving frame rather than sample to not disturb the latter, insulating the electronics from the sample environment, distributing the light with polymer waveguides, computing focus metrics on-the-fly) and complex devices (embedded FPGA with multiple detectors, custom detection lenses). It successfully overcomes the shortcomings of traditional commercial microscopes for this particular application (imaging speed, stability, throughput). While I am not equipped to judge the clinical impact, the authors convincingly demonstrate that their microscope can turn tedious and lengthy studies into more feasible tasks.

Overall, I would have hoped to find more quantitative measurements of the instrument capacities and limitations (e.g. temporal and spatial resolution). Additionally, the manuscript would have a much higher impact would the design files be open to the community, thus allowing for reproducibility of the research. Yet, I understand that the instrument has some commercial potential and that a patent application has been submitted.

In my opinion, the manuscript is conform to Nature Communications publication criteria and should be accepted with minor changes. I have organized my comments between major (important comments I would like to see addressed) and minor comments (small remarks).

We wish to thank the reviewer for the very kind words and the positive judgement of our system. Regarding the instrument capabilities, we have added substantial new information (see specific responses below) that we hope are able to satisfy the very legitimate inquiries. Regarding the design files: the system has been developed over many years and hence is highly complex concerning hardware-software-codesign and fabrication/assembly/calibration of modules. It now consists of dozens of custom designed non-standard parts (like e.g. the objectives or the board for the FPGA and CMOS chips), covers very specific code/software for FPGAs, embedded controllers and the main PC. The hardware cost alone amounts to approximately 50,000.- €. We provide a 2.5 page description of the setup in the accompanying methods section containing technical specifications in high detail. In addition, the figures show

high-resolution photographs and schematic drawings of the key components and the system logic. But the system was developed based on a wide range of expertise and know-how in the fields of microelectronics, circuitry, optics, cell-imaging, mechatronics, informatics and artificial intelligence. Therefore, an uncontrolled public dissemination of detailed system knowledge would not be acceptable. We hope that the reviewer acknowledges this as the most detail we can offer at the moment without compromising our future plans for commercialization. However, we would be more than happy to open the system to interested collaborators upon request.

Major comments:

1. The authors state in the results that “The resolution was numerically ~74% that of the standard system (2.1 px/μm vs. 1.55 px/μm in ComplexEye).” In this case this is the pixel resolution (akin to magnification) and should not be mistaken for optical resolution. I encourage the authors to make this point clear to avoid misunderstanding. The authors should also compute the magnification of their custom optics from the pixel resolution and indicate it, in order to help readers better grasp how it compares with the commercial instrument and its 20x lens. In addition to magnification (as they are related), the authors should also mention the total field of view size (pixel resolution x sensor size).

The reviewer is right and we apologize for our incorrect description. The optical resolution of our system is exclusively defined by the NA of the optics, which is 0.3 for ComplexEye. The optical resolution for objects illuminated with white light is $\lambda/(2 \times \text{NA})$ (where λ is the wavelength in nm (in white light assumed to be 550 as median value)), hence 917 nm. In the Leica system we used a 20x lens with a NA of 0.4, hence an optical resolution of 688 nm. Thus, the Leica is indeed providing a 25% better resolution, than the ComplexEye, but this cannot be seen from the pixel resolution. The differences in pixel resolutions of the Leica system and ComplexEye result from the different imaging chips. In the Leica this is a Sony ICX285 monochrome with 1.392x1.040 pixels (pixel size 6.45x6.45 μm), in ComplexEye this is an OmniVision OV9715 color chip with 1.280x800 pixels (pixel size 3x3 μm). From the pixel resolution of 0.64/μm we can compute a FOV of 825.8x512 μm for ComplexEye. In the Leica (pixel resolution 0.48/μm) the FOV is 662.9x495.2 μm. The magnification of the ComplexEye optics is thus 4.69 (pixel size/pixel resolution = 3/0.64), while for the Leica this is 13.43 (6.45/0.48). We have now exchanged the images in Fig. 1c to show the entire FOV and indicate the size of the FOV rather than the pixel numbers per 250 μm grid in a Neubauer chamber. In addition, we have updated the MS text in the first paragraph in results. **The text now reads: “The optical resolution of our system is exclusively defined by the NA of the optics, which is 0.3 for ComplexEye which leads to 917 nm at 550nm illumination. In the Leica system we used a 20x lens with a NA of 0.4, hence an optical resolution of 688 nm. Next, a Neubauer counting chamber was used to**

acquire images with a conventional microscope and our array system to determine the total FOV. (Fig. 1c). Thereby, the conventional Leica system demonstrated a FOV of 662.9x495.2 μm and the ComplexEye 825.8x512 μm . The magnification of the ComplexEye optics is thus 4.69, while for the Leica this is 13.43.”

Figure 1 c, Images acquired with a standard 20x lens (left) and the ComplexEye lens (right) of a 250 μm grid in a Neubauer chamber. The size of the FOV in the respective captured images are displayed.

2. Figure 1c is cited as showing that the microscope leads to image that are “optically similar” to that of a commercial instrument. The images in 1c only show that the two systems have different magnifications, and even that fact is not self-evident as the two images are scaled to similar size and only the number of pixels gives it away. The authors should use a USAF pattern (probably a negative one for bright field, see Resolution Test Targets in Thorlabs for examples) as this would constitute a much better comparison of optical resolution/quality. In addition, using a USAF pattern would allow them to quantify the contrast in different parts of the field of view, which could be a Supplementary figure.

We thank the reviewer for this important and constructive comment. We ordered positive 1951 USAF Wheel Pattern Test Targets and with these quantified the contrast in different parts of the field of view. We also made the same images with the Leica system for the reader to compare. From these images we produced a new **Supplementary Fig. 1**. The image is reproduced below for the reviewer. We would still like to keep the images of cells in the main figure of the MS, since it shows the optical performance of the system directly on a relevant biological sample. Furthermore, we added this information to the MS (first paragraph of results). The text now reads: “Connected to a megapixel sensor these lenses generate images that are optically comparable to those made with a conventional microscope (Fig. 1b, Supplementary Fig. 1) and, unlike images in a recently published array microscope, do not require computation-intensive postprocessing. This was verified by imaging USAF Patterns to compare the optical resolution in different parts of the field of view (FOV) of the ComplexEye and a commercial microscope (Supplemental fig. 1).”

Supplementary Fig. 1 | **Comparison of optical resolution.** To compare the optical resolution between ComplexEye and Leica and also to quantify the contrast in different parts of the field of view of both microscopes, a positive 1951 USAF Wheel Pattern Test Target (R3L1S4P, Thorlabs) was imaged. The smallest pattern was imaged at 5 different areas of the FOV (middle, top left, top right, bottom left and bottom right). Here it was shown that both

microscopes were able to resolve the smallest pattern with a resolution of 228 line pairs per millimetre (lp/mm) shown in group 7/ element 6. However, there are slight blurs in the upper parts of the top left/right fields in ComplexEye, that are absent in the Leica system pointing towards a not 100% co-aligned surface of the imaging chip with the focus area of the lens in our detection board.

3. The temporal resolution of the experiments is a key consideration in this study, as the authors insist on the 1 image / 8 seconds (per well) limit under which neutrophils can effectively be tracked. Without clear breakdown of the different steps of the imaging in terms of time (stage travel speed, exposure time, refocusing), it is difficult to follow the comparisons with previous work and to easily grasp the temporal limitations. For instance, the authors write in the introduction: "The staggered lenses required 8 steps to image all wells of a 96-well plate[31], still double the time required for immune cell imaging in 96-well plates and 8-times too slow for 384-well plates.". This statement doesn't make sense without telling the reader how long each of the steps took for imaging. If each of the 8 steps took one second only, it could be sufficient to image a 96 well plate at the desired temporal resolution. The other aspect of this statement that I find troubling, is that it seems to imply that the other published instrument is capable of imaging 48 wells of a 96 well plate at the required speed (since it images the whole plate 2x too slow, it can image half of it at the required speed), well over the authors' system (16 for a 96 well plate) if we are to believe the following sentence a paragraph later: "ComplexEye [...] to simultaneously image 16 adjacent wells of a 96 well plate or 64 wells of a 384-well plate with one frame per 8 seconds". Likewise, the imaging of a 384 well plate by the other published microscope is allegedly 8 times too slow, meaning that it could potentially also image 48 wells of a 384 well plate within the required speed. However, the well to well distance in a 96 well-plate is twice that of a 384 well-plate. Therefore if stage travel time is a major factor limiting the imaging speed, they should be able to image more regions in the 384 well-plate. Without additional explanation, the statement could be considered misleading. Finally, I am puzzled why ComplexEye seems to be limited to imaging only 16 of the 96 wells on a 96 well plate at the necessary imaging speed? Is it an actual limitation or simply the maximum imaging speed used by the authors in the present manuscript?

All in all, the authors should clarify these points and discuss in more details the typical exposure time used with such samples, the time needed for refocusing, the potential need for a rest period after moving the stage (if applicable) and the typical stage travel speed considered "safe" for imaging neutrophiles. This would help clear potential confusion and put in perspective the numbers (1 frame / 8 seconds, number of wells that can be imaged). An additional figure

illustrating the factor limiting spatio-temporal resolutions (typically stage travel speed, refocusing) could be useful and improve the reading experience.

We apologize for having been not clear enough in this introductory part. The 12-channel microscope assembled by Cribb et al. is designed for measuring beads on the surface of cancer cells which they image for 60 s before moving on to the next 12 positions. Hence, they would image all wells of a 96-well plate within 8 minutes. This is by design, as the required readout (bead micromovements on cells) is very different from our setup. Therefore, the systems cannot be directly compared. However, the system features an electrical XY-stage for the movement of the 96-well plate against a stable set of 12 objectives. In that sense it is not different from a standard single lens system with an electrical stage. When imaging non-adherent cells like neutrophils, we have found that moving the stage faster than required to visit 4 positions in 8 seconds (2 seconds per position), the required acceleration and deceleration steps of the electrical stage become so aggressive, that the neutrophils show movement artefacts not brought about by autonomous migration. This is the reason, why the published 12-channel system of Cribb et al could image max. 48 wells of a 96-well plate per 8 seconds to image non-adherent cells. Even though the distance of wells in a 384-well is just half that of a 96-well, still aggressive acceleration- and deceleration steps would be required to image cells fast enough. So, in general, microscopes that move the plate rather than the optics quickly reach limits of imaging non-adherent cells that are not defined by available stage travel speeds. Based on this comment, we have updated the introduction of the MS text. The text now reads: "This 12-channel microscope features an electrical XY-stage for the movement of the 96-well plate against a stable set of 12 objectives. In that sense it is not different from a standard single lens system with an electrical stage and moving the stage too quickly would also cause movement artefacts. Therefore, the 12-channel system of Cribb et al. could image maximum 48 wells of a 96-well plate per 8 seconds to study non-adherent cells. In general, microscopes that move the plate quickly reach limits of imaging non-adherent cells.

...

ComplexEye is different by design, as it moves the optics against a stable plate. Hence, stage travel speed can be increased to boost the throughput of the system without having to fear movement artefacts in non-adherent cells." We currently feature illumination times of 1/30 second, XY-travel speeds of 10 mm/sec and 20 mm/sec of the Z-drive for focussing. One entire process of positioning, focussing and imaging is done in under 2 seconds allowing to visit 4 positions per lens, before the first position has to be imaged again. It would be possible to speed this up by a factor of 4 for imaging 1,536 well plates, but this would require the implementation of faster XY-tables and control hardware. However, with the current setup it would not be possible to image e.g. 32 wells of a 96-well plate, as the travel speed and reach

of the XY-stage does not allow a travel of 40 mm to arrive at the next group of 16 wells fast enough to stay within the 8 seconds/frame limit. Hence, with a 16-lens setup it is not possible to image more than 16 wells in a 96 well or 64 wells in a 384-well plate.

We have updated the MS text at the part describing fig. 2 with this information. The text now reads: "ComplexEye features illumination times of 1/30 second, XY-travel speeds of 10mm/sec and 20mm/sec of the Z-drive for focusing. One entire process of positioning, focusing and imaging is done in under 2 seconds allowing to visit 4 positions per lens, before the first position has to be imaged again (Supplementary Fig. 3)".

In addition, we have taken the suggestion of the reviewer and produced a new Supplementary Fig. 3 that illustrates these features (reproduced for the reviewer below).

Supplementary Fig. 3 | **Mode-related focusing speed considerations.** Autofocus variants of one focusing phase regarding number of video channels and autofocus mode. While in free running collective autofocus mode the drive speed of the Z-stage is independent of the channel count (left two diagrams). In the focus position map-based triggered mode (right two diagrams) each focus level requires a rest time of ~40 ms and in conjunction with acceleration and

deceleration times of the stage overruns the 8 s limit in a 96 video channel constellation (lower right diagram). Hence, a 96-lens setup would only be possible with the free-running autofocus, if staying in the 8 seconds/frame limit is required.

4. The authors state that the use of white LED allows “generating a homogeneous illumination (Fig. 1d)”. But Figure 1b clearly has a background gradient. The authors should provide a Supplementary figure showing a side-view intensity plot of the illumination and justify the “homogeneity” statement quantitatively. Is this pattern similar in all 16 illuminations?

The reviewer is correct. We do have a slight background gradient. This is due to small imperfections of the internal construction of the objectives leading to weak reflections. Despite intensive optimization (careful internal blackening, using rough internal surfaces, optimizing the bearings of the glass elements) we could not completely eliminate these reflections, but they do not interfere with cell tracking. To still address the reviewer’s point we now imaged an empty 96-well plate focused on the bottom and took images of 16 wells with the 16 illuminations. As a comparison we also took a single image from one well with the conventional Leica microscope. Afterwards, the grey values of the images in horizontal and vertical lines were evaluated with Fiji and an intensity plot for the single illumination and 16 illuminations together was generated. We have added a new Supplementary Fig. 2 with these data and reproduced them below for the reviewer. In addition, we have also added this information to the MS (description of Fig 1d in results). **The text now reads: “In a suitable array microscope each lens requires its own Köhler-optimized illumination to achieve perfect image quality. We solved this problem with a white LED whose light was distributed by polymer fibers to each of 16 focusing elements, generating a homogenous illumination (Fig. 1d, Supplementary Fig. 2).”**

Supplementary Fig. 2 | **Illumination homogeneity**. To demonstrate the homogeneity of all 16 illuminations of the ComplexEye and to relate it to the conventional Leica microscope, the bottom of an empty 96-well plate was focussed and imaged. The grey values of the images in horizontal and vertical lines were evaluated with Fiji and an intensity plot for the illumination was generated. For ComplexEye the intensity plot of a single illumination and of all 16 illuminations is shown in the upper part. For the vertical illumination, there was a slight gradient on the right side. Besides this, ComplexEye displayed a homogeneous illumination. The data for the 16 illuminations are shown as mean + StDev. (grey area) of all 16 illuminations.

5. The following sentence is complex and not entirely clear: "the whole 16-fold lens unit was moved periodically up and down within the overall focus range while in every round each of the 16 imagers took a picture in the moment of maximum image sharpness at the predefined frame to frame time-gap of the resulting Movie". The most logical deduction to me is that for

each well, the authors calibrated before imaging an optimum focus for the corresponding lens. For each position of the lenses array, the distribution of focus positions among the lenses defines a focus range. Then, during imaging, at each position, the stage goes through that range and the sensors only take pictures at the calibrated focus position of their corresponding lens. This interpretation is not compatible with the “Data handling” part of the methods, which seems to imply that images are recorded at every focus position and only the maximum sharpness image is kept in the FPGA. The authors should make their statement clearer. A Supplementary figure illustrating the imaging protocol would be very useful. Finally, in order to satisfy the more optics-minded readers, the authors should provide a Supplementary figure showing an example of focus distribution between lenses (e.g. stage height distribution while imaging the same well with each lens) and across wells (e.g. imaging all wells with a single lens and plotting the stage z position distribution).

We are sorry for the confusion and understand the seemingly incompatible interpretations very well but both interpretations are correct. The system has two modes of operation, the *free running collective autofocus mode* and the *focus position map based triggered mode*. In the free running collective autofocus mode all 16 imager-chips run when started at maximum frame rate (30 FPS @ 720p resolution) for the time duration of one focusing period (e.g. approx. 2 s for 64 wells with the 16-fold system) and deliver data of each frame to the FPGA. The FPGA calculates a sharpness indicator value for each frame of each well, compares the value to the frontrunner of the last frames of the same well and stores the frame-data in case of a larger value for that well. During this free running data acquisition process within one focusing period the Z-stage of the microscope unit is raised or lowered at constant speed in Z-direction within the focal range of all 16 lenses (approximately 80 to 150 μm depending on well plate tolerances, see Supplementary Fig. 3 below) and therefore realizes a collective focussing. This leads to one sharpest image of all acquired images of all 16 imagers at the end of each focusing period. Unfortunately, the spread of the focal range over all lenses was nearly 4 times larger than expected (which is based on the imprecise height per well in available standard 96-well plates, see image below) which resulted in a sharpness blur in the video (varying from frame to frame). This blurry variation, although having no effect on the overall tracking results, is irritating and disturbing for viewers of the videos. Therefore, we have decided to use a *focus position map based triggered mode* in the current system constellation, in which these effects do not occur. In the *focus position map based triggered mode* each acquisition cycle of all 16 imagers as a single shot action is triggered by a command that the PC software sends to the FPGA that controls the imagers. Prior to the video generation in an initialization sequence the PC-software uses this channel-wise triggered control mechanism to automatically detect and store the focus position (z-position) of all lenses using a successive approximation technique. Afterwards the list of all focus position numbers together with their corresponding channel

numbers are sorted by ascending Z-value. During the following periodical video frame acquisition the PC directs the Z-stage to move the microscope unit to each focus position of the list, triggers the FPGA to take an image of the corresponding imager channel and repeats this procedure for all 16 imagers. This operating mode results in flicker-free videos, but is speed-limited and does not offer the ability to dynamically adjust focus during long video recordings. It would be possible to overcome these limitations with the next generation imager units and electronics that will allow us to return to the more innovative and powerful *free running collective autofocus mode*.

To determine the focus distribution between the 16 lenses of the ComplexEye, we performed an experiment, where 50 μ l of heparin beads were pipetted into 16 wells of the 96 well plate and imaged with the ComplexEye. Heparin beads were used as they have a distinct shape and quickly sink to the bottom of the wells. In all wells, the beads were focussed, the Z position of the best focus was evaluated and a picture of the focussed beads was taken (see figure below). This experiment served to show that there is a substantial spread of Z in the 16 wells.

Figure for Review only:

Focus distribution between lenses in a standard 96-well plate

a, To display the focus distribution between the 16 lenses of the ComplexEye when imaging a standard 96-well flat bottom plate, 50 µl of heparin beads were pipetted into 16 wells of the 96-well plate. Heparin beads were used as they have a distinct shape and quickly sink to the bottom of the wells. In all wells, the beads were focussed, the Z position of the best focus was evaluated and a picture of the focussed beads was taken. **b**, 16 images of beads taken with the single lenses. Thereby, the Z value of well B04 was artificially set to 0.00. The numbers on the images display the deviation of the Z position from the B4 value in mm. Note that the entire spread of Z in these 16 wells is 220 µm (from 0.15 to -0.07)

6. I did not find the “Data availability statement” in the manuscript. The only dataset provided is a very small example used to illustrate the pipeline. Given the argued impact of the high-throughput study, the authors should provide the raw images and scripts to allow reproducing their results, for instance by depositing them on Zenodo.

We agree with the reviewer and have now uploaded all Movies of round 8 of the compound screen on Zenodo. To reproduce our results, raw data of videos can be found at <https://zenodo.org/record/7962145> (full dataset upon paper acceptance).

Minor comments:

1. What is the power of the illumination used on each illuminated area, or better the illumination power per square centimeters?

We have measured the illuminance in the circular illuminated field of each well with a diameter of about 2 mm. It has a level of ~6,500 lx at 12.5% of the maximum illumination system power output, a value that is typically used for video recording. Assuming a reference wavelength of 555 nm, a light power of about 952 nW/cm² can be derived from this, or a power related to the illuminated area (0.031 cm²) of about 30 nW. We have added this information in the main text at the description of fig 1d: “Thereby, the power of illumination related to the illuminated area (0.031 cm²) was measured and determined as 30 nW.”

2. Figure 1b is missing scale bars.

We have now added a scale bar in 1b (see below) and updated figure 1 in the main MS.

3. As the authors state: “To protect the electronics from the humidity in the incubator, we encapsulated them to separate them from the sample level (Fig. 2c)”. The authors could specify by which means does the encapsulation protect the electronics from humidity (sealing?). How is dryness maintained in the box, simply by insulation?

The system design provides for the separation between the humid area (cell environment) and the dry area (technical environment) through encapsulation measures and suitable start-up

procedures. In short, the mechanical X/Y/Z stages (to prevent corrosion and maintain adequate gliding properties of the lubricant) and the electronics (to prevent corrosion, short circuits and operating point shifts of components) must be protected from moisture. The sealing measures and concepts for this are described in the methods section. The necessary start-up procedure of the system during a cold start first provides for a temperature increase in the incubator to the target temperature of the measurement (37°C) with the well plate airlock open, in order to quickly bring all components of the system to the same temperature before activating the humidity and gas regulation, and to ensure uniform humidity and pressure distribution inside the incubator. Afterwards, either the well plate airlock is closed as described in the text or a well plate is inserted and locked. Only then are the humidity and gas regulation activated. Any residual moisture caused by minor leaks or improper operating procedures can additionally be reabsorbed via exchangeable silica gel pads. Importantly, all experiments and metrological issues addressed with the system on cell samples in this study could be performed during video recording time periods (normally one to four hours) that did not require a humid environment or a CO₂ enriched atmosphere.

4. “The analysis showed that the measured values were comparable to our migration data obtained before”. Could the author explicitly state the values obtained in their previous work? And potentially make them apparent in figure 3a.

In 2022 Langer et al.³ have analysed the migration patterns of patients with cirrhosis and compared the results with 25 healthy individuals. We have now added the migration values of these 25 individuals to figure 3a (orange lines) and updated the figure in the main MS. The comparison with the reference migration values shows, that our results in figure 3a are similar.

5. On Fig 3, the authors should mention the number of tracks per points on average (and potentially with a standard deviation).

The tracks are divided in total and valid tracks. Thereby, valid tracks are determined by two parameters: minimum track duration and movement threshold. For neutrophils, the minimum track duration is defined as 1 min and the movement threshold 8 µm (one cell diameter). As neutrophils treated with PBS and neutrophils stimulated with fMLP, CXCL1 and CXCL8 display

completely different numbers of tracks, we wanted to demonstrate the number of total and valid tracks and the mean track duration separately for all conditions. For simplicity, we have presented the data of all experiments from figure 3 in the table below.

Parameter	PBS	fMLP	CXCL1	CXCL8
Total tracks mean	384	1260	365	798
Total tracks Std. deviation	117.7	413.5	72.5	174.6
Valid tracks mean	238	881	231	572
Valid tracks Std. deviation	70.3	278.8	56.2	168.5
Mean track duration [min]	40.9	11.4	40.7	15.9
Mean track duration [min] Std. deviation	14.8	3.3	11.9	5.2

6. Figure panels 4a and 4b are too small, I would advise the authors to split the figure into two to allow more comfortable reading. Consequently, this would lead to less crowded figures

We thank the reviewer for this suggestion. If the editors approve it we would split figure 4 into two new figures as suggested below:

Figure 4 | ComplexEye high throughput screening of migration modifying compounds.
a, Experimental setup of the screening assay. Briefly, neutrophils from human blood were isolated and plated on a 384-well plate, treated with one of the 1,000 compounds from a library of known bioactives and stimulated with fMLP. Neutrophil motility was then recorded simultaneously in 64 wells of a 384-well plate for one hour (8 seconds between frames) using ComplexEye. Afterwards the motility was analyzed via single cell tracking. **b**, Data represent 1,000 Movies, ~800 tracks/Movie and show the impact of 1,000 compounds screened in 17 rounds, each round with three controls (PBS, DMSO and fMLP or fMLP/DMSO). The heatmap shows each round with 64-wells with the relative speed of imaged neutrophils indicated as color code compared to the fMLP-control in that run (artificially set to 1.0). Compounds that

reduced the speed are shown in green-blue (low speed). Indicated grey wells were non-evaluative due to production residues of the 384-well plates inhibiting clear sight of the cells.

Figure 5 | ComplexEye high throughput screening identifies neutrophil migration modifiers. **a**, Relative speed data with 17 compounds reducing motility by more than 40% compared to fMLP as detected by the screen illustrated in Fig. 4. Compound R14D2 had less effects on speed, but strongly affected the cell shape. **b**, Relative activity data with 27 compounds reducing the number of migrating neutrophils by more than 20% compared to fMLP. **c**, Sorting of inhibitory compounds into classes according to their effect on speed and activity of migrating neutrophils. In every square the left vertical line is for relative speed and the right vertical line is for relative activity. Class 1: compounds strongly decreasing the speed whereas the number of migrating cells was not affected. Class 2: compounds strongly decreasing the number of moving cells without affecting their speed. Class 3: compounds decreasing both, speed and activity. **d**, comparison of neutrophil morphology between fMLP-treated cells and cells treated with fMLP and the indicated compounds. Polar plots show migration tracks of all cells in the experiment normalized to one common center. Rings in polar plots define 100 μ m distances.

7. The authors should provide a rough estimate of the price of their custom instrument.

The procurement costs of all components, semi-finished products and functional assemblies for the 16-fold system amount to approximately € 50,000. The largest items are the lenses at around €12,000 and the high-precision stage system at €18,000. For a 96-fold system, which includes 6 of the 16-fold camera clusters shown in this study, plain hardware costs of about 100,000 € have to be expected.

Reference for Reviewer #2:

- 3 Langer, M. M. *et al.* Pathological neutrophil migration predicts adverse outcomes in hospitalized patients with liver cirrhosis. *Liver Int*, doi:10.1111/liv.15486 (2022).

REVIEWER COMMENTS

Reviewer #1 (Remarks to the Author)

The comprehensive responses addressed certain worries raised by both reviewers, and I concur that the ComplexEye displays innovation in multiple dimensions. Nevertheless, there remains concerns surrounding the technology's potential to revolutionize microscopy at its core or to notably propel our comprehension of cell migration. As concurred by the authors themselves, the analysis of fluorescence holds critical significance within microscopy, particularly for clinical specimens encompassing a mix of diverse cell types, as it serves to differentiate subtypes of migrating immune cells.

While the authors extensively compared their system's field of view (FOV) with that of the ECLIPSE Ti2 system, it's important to note that if fluorescence is employed for cell tracking, the images captured by the ECLIPSE Ti2's 4X objective lens are capable of reliably tracking individual cells. Despite the apparent cost-effectiveness of the ComplexEye system in contrast to the ECLIPSE Ti2, the final sales price of a microscope must encompass a multitude of expenses, including administrative, marketing, sales, and service costs. Consequently, the pricing for marketing a ComplexEye system may not differ significantly from that of existing commercial microscopes. Considering all aspects including fluorescence imaging capability, throughput, and cost, the ComplexEye lacks a substantial advantage over conventional microscopes.

Relying solely on in vitro migration assays doesn't carry substantial weight in terms of biological or clinical relevance. The fact that certain compounds exhibited inhibition of cell migration in vitro does not necessarily imply congruent effects in an in vivo context, let alone potential treatment efficacy. Within the new dataset validating the compounds, two concerns emerge: firstly, the rationale behind the selection of 8 out of 17 migration-inhibiting compounds; secondly, the failure to reproduce the results for 3 out of the tested 8 compounds. Notably significant discrepancies exist between the screening and validation values for several compounds, casting doubt on the reliability of the screening experiment.

All these facets collectively dampen the enthusiasm for this study. Considering the technological innovation alongside limitations in biological relevance and assay effectiveness, it is advised that the paper finds publication in a more specialized technology-focused journal, as opposed to the broader multidisciplinary scope of Nature Communications.

Reviewer #2 (Remarks to the Author)

The authors answered most point raised during the review adequately. In particular, they performed various experiments that clarified interrogations regarding the timing and optical capacity of their microscope, modified their manuscript accordingly and clarified certain statements in the text.

I am left with two comments, one that I considered major for the purpose of making the paper clearer (No 1), which follows from the rebuttal, and another comment that is minor (NO 2) and corresponds to a comment raised during the first review cycle.

Comment No 1 (major)

Following the authors' rebuttal for major comments 3 (temporal breakdown of acquisition steps) and 5 (confusion regarding the axial focusing), they added Supplementary Figure 3. While the figure is informative, it is not sufficient by itself to clear the confusion about the axial focusing:

- The method section should contain the description of the two axial focusing modes with reference to Supplementary Figure 3 (description that already exists in the authors' rebuttal).
- Line 217 should refer to the name of the axial focusing mode used and point to the method for details.
- Line 506 should describe what happens to the on-the-fly processing for each axial focusing mode (clarify what happens in the FPGA in each case).
- In the caption of Supplementary Figure 3, the sentence "Autofocus variants of one focusing phase regarding number of video channels and autofocus mode" is complex and not clear to me. Additionally, the name of the two different autofocus modes could be highlighted to improve reading (e.g. quotation marks or bold).

Comment No 2 (minor)

The authors described a rough breakdown of their cost in the rebuttal (minor comment 7), but while this is interesting for the reviewers, it is an information rather aimed at the readers. I still suggest adding the financial breakdown in the methods of the paper.

I can only assume that the future commercialization of the microscope is the reason for not writing it out directly into the paper. I would then suggest to add something along the line of "This price is a rough estimate that does not take into account price fluctuations, or the development, assembling, and personal costs." I leave the ultimate decision to include or not this information in the manuscript to the authors.

Decision

I believe that the manuscript should be accepted with minor changes corresponding to the comment No 2 (major).

Responses to reviewers for “ComplexEye - a multi lens array microscope for High-Throughput embedded immune cell migration analysis” by Cibir et al.

Response to Reviewer 1.....Page 2
Response to Reviewer 2.....Page 7

Reviewer 1:

The comprehensive responses addressed certain worries raised by both reviewers, and I concur that the ComplexEye displays innovation in multiple dimensions. Nevertheless, there remains concerns surrounding the technology's potential to revolutionize microscopy at its core or to notably propel our comprehension of cell migration. As concurred by the authors themselves, the analysis of fluorescence holds critical significance within microscopy, particularly for clinical specimens encompassing a mix of diverse cell types, as it serves to differentiate subtypes of migrating immune cells.

In our response to the first review, we outlined that our system is already very powerful even without fluorescence, and we demonstrated this through extensive measurements including a migration screen of fast-moving neutrophils with 1,000 substances, which would simply not be possible with conventional systems. Even though fluorescence would be a useful addition to the system, we have demonstrated that very important new applications can be realized without it. Measuring mixtures of cells is in itself complicated and would require massive efforts for standardization. In contrast, any approach targeting at the modification of a specific cell type like, in our case, neutrophils, would investigate the purified cells first, as we have done. Hence, the current lack of fluorescence is not inhibiting ComplexEye to make fundamental and previously impossible contributions to the field. If we take an example from consumer electronics: when the first iPhone was introduced in 2007, it had a small screen, poor camera, small memory etc. Still it completely revolutionized communication. Present day iPhones are multiple times more powerful and nobody would invest in a first-generation device any more. But the fact that the first iPhone was made available started a whole industry. In that sense we consider ComplexEye as the first of its kind with the capability of revolutionizing the field of high-throughput video-microscopy. Only the response of the scientific community will be able to show, what its impact is beyond measuring the migration of neutrophils.

While the authors extensively compared their system's field of view (FOV) with that of the ECLIPSE Ti2 system, it's important to note that if fluorescence is employed for cell tracking, the images captured by the ECLIPSE Ti2's 4X objective lens are capable of reliably tracking individual cells.

We wish to clarify again, that a single lens system, including the ECLIPSE, is unable to perform the video recordings of migrating immune cells in the same throughput, like the ComplexEye can. The main reason for this is the requirement to move well after well of a 96-well plate in front of the lens for picture generation in the ECLIPSE. In order to obtain useful video sequences of migrating neutrophils this has to be repeated once every 8 seconds minimum. For reasons explained in detail in the manuscript (shaking artefacts prominent in non-adherent cells when the table moves too fast) these boundary conditions limit a single lens system to 4

wells of continuous recording, which we have verified extensively based on own experimental results with a single lens system. Using a 4x lens does not change this fundamental limitation, as a 4x lens would still see only the major parts of one whole well of a 96-well plate, but not more than one (e.g. two adjacent wells). Hence, also with a 4x lens still the plate would have to be moved, thus inducing the said artefacts. Only the ComplexEye setup with one lens per well and 16 running simultaneously is able to generate 16 or 64 movies without artefacts as we demonstrate. In ComplexEye, movement artefacts are further inhibited by moving the lens against a fixed table, which completely eliminates the need for moving the shaking-sensitive sample. Furthermore, suggesting the use of a 4x lens on an ECLIPSE ignores the fact, that the cells would be extremely small in such recordings and not show much details of cell shape, which, however, are very important for the analysis, as we have shown. Taken together, ComplexEye is fundamentally different from conventional single-lens systems like the ECLIPSE.

Despite the apparent cost-effectiveness of the ComplexEye system in contrast to the ECLIPSE Ti2, the final sales price of a microscope must encompass a multitude of expenses, including administrative, marketing, sales, and service costs. Consequently, the pricing for marketing a ComplexEye system may not differ significantly from that of existing commercial microscopes. Considering all aspects including fluorescence imaging capability, throughput, and cost, the ComplexEye lacks a substantial advantage over conventional microscopes.

Our manuscript is not about a marketable product. As requested by reviewer #2, we have now provided rough estimates of the hardware costs associated with a system (new paragraph at the end of the discussion), but we are not in a position, nor have we claimed to be, to offer a product that can be purchased. Also, we never claimed, that ComplexEye would offer cost advantages over existing systems. What we do claim, and this is valid and proven in our MS, is that ComplexEye can perform measurements that are not possible with any commercial system currently available. Whether this might finally end up in a marketable product and how competitive this might possibly be is unclear. However, we feel that such considerations should not be relevant in a basic science manuscript.

Relying solely on in vitro migration assays doesn't carry substantial weight in terms of biological or clinical relevance. The fact that certain compounds exhibited inhibition of cell migration in vitro does not necessarily imply congruent effects in an in vivo context, let alone potential treatment efficacy. Within the new dataset validating the compounds, two concerns emerge: firstly, the rationale behind the selection of 8 out of 17 migration-inhibiting compounds; secondly, the failure to reproduce the results for 3 out of the tested 8 compounds. Notably significant discrepancies exist between the screening and validation values for several compounds, casting doubt on the reliability of the screening experiment.

It is important to clarify, that our MS is about a new concept for a fast video microscope with the search for modulators of Neutrophil migration being a novel and important use case where the system demonstrated its unprecedented performance. With this screen as an example, researchers in the field can now evaluate, what they would do with such a machine and we truly hope that our contribution can kick-start a whole new line of research. However, since it is not the main thrust of the MS to follow up on the performance of newly identified migration modulators in vivo, it would be way beyond the scope of this MS to engage in a series of experiments with disease models or even clinical trials in humans. This will be the focus of independent studies, as the development of suitable animal models and the ethical approval for such studies require a time frame in the order of years.

We understand, however, the reviewer's concern that only 8 out of 17 compounds were selected for validation. At the time point of the first revision we had ordered all 17 compounds for validation experiments. Unfortunately, there were substantial problems with the availability of 9 out of 17 compounds. Therefore, we did the first round of validation with only 8 compounds to demonstrate the principal validity of our screening results. As a response to this new request and since the remaining 9 compounds have meanwhile arrived we now also tested them in a new set of experiments with freshly isolated neutrophils from two healthy volunteers (see figure below). Here, the migration-decreasing capacity of 7 of the 9 compounds was confirmed, while the other two compounds showed slight differences in their migration-reducing capacity compared to the initial screen. In the figure below (for review only) we put the results from the first validation round (a) with 8 compounds and the actual validation round (b) with 9 compounds together. With these new experiments all 17 hit compounds from the first round were validated with neutrophils from two additional individuals (hence three in total). We can thus confirm 12 of 17 compounds, which corresponds to a 71% confirmation rate of the results of the ComplexEye primary screen.

In the light of these results we do not agree with the reviewer's view that the first screening could not be reproduced in a second round and this would indicate the unreliability of our screening approach in general. We would like to remind the reviewer that developing a new drug and bringing it to the market is a long and difficult process. It begins with drug discovery: the unearthing of promising compounds which demonstrate a desired biological effect (in our case; reduction of fMLP-induced migration in human neutrophils). Compound screening is a key method by which initial drug discovery can be carried out. Here, large numbers of compounds are tested in high throughput. In these initial screens, compounds are typically tested in single concentrations and as singlets. In this way, compound screening enables those compounds which produce desired effects – known as “hits” – to be identified. Once this is done, the process of further testing and drug development begins, whereby validation of initial

hits with an increased number of biological replicates (n) is one of the first steps. Here, it is completely normal, that a hit compound found in a primary screen cannot be validated in a secondary. This is exactly the rationale for doing validation screens. At the beginning of every drug development process one starts with many hits and ends up with a handful of confirmed ones. Therefore, we cannot understand the criticism regarding the outcome of our validation experiments and the assumption derived from it, that our screen would be unreliable. In our experiments, the hit rate in the primary screen was 1.7% (17 out of 1,000 compounds). The validation rate of these hit compounds was 71% (12 out of 17 compounds). These rates are well within the normal range of drug screenings. To give an example, Gómara-Lomero et al. performed in vitro synergy screens of FDA-approved drugs against *Klebsiella pneumonia* and came up with initial hit rates ranging from 2.2% to 2.9% and validation rates of 15.1% to 65.9%¹.

Figure for review only:

Validation of hit compounds

To determine, whether the speed-decreasing compounds identified in the first round of the screening would show the same effect in additional experiments with neutrophils from different donors, the compounds were tested in two additional rounds with neutrophils from two different healthy volunteers. **a**, In a first round (first revision), 8 out of 17 compounds were validated. Here, the migration-reducing capacity of 5 of the 8 compounds could be confirmed. **b**, In a second round (new data, this revision), the remaining 9 of the 17 compounds were validated. Here, 7 of the 9 substances could be confirmed as migration-decreasing compounds. Black

circles show the % speed reduction of n=2 healthy volunteers and green points display the level of speed reduction of the same compound as found in the initial screen. The red dashed line indicates 40% speed reduction.

An information about these results has been added to the last chapter of the results. It reads:

“Of the 17 compounds identified in the first screen 12 (71%) could be validated for their function in a second analysis (data not shown). Future studies will now allow to investigate the underlying cellular mechanisms in great detail and hence arrive at a completely new understanding of fMLP-induced neutrophil motility.”

Reference for Reviewer 1:

- 1 Gómara-Lomero, M., López-Calleja, A. I., Rezusta, A., Aínsa, J. A. & Ramón-García, S. In vitro synergy screens of FDA-approved drugs reveal novel zidovudine- and azithromycin-based combinations with last-line antibiotics against *Klebsiella pneumoniae*. *Sci Rep* **13**, 14429 (2023). <https://doi.org:10.1038/s41598-023-39647-9>

Reviewer 2:

The authors answered most point raised during the review adequately. In particular, they performed various experiments that clarified interrogations regarding the timing and optical capacity of their microscope, modified their manuscript accordingly and clarified certain statements in the text.

I am left with two comments, one that I considered major for the purpose of making the paper clearer (No 1), which follows from the rebuttal, and another comment that is minor (N0 2) and corresponds to a comment raised during the first review cycle.

Major:

Following the authors' rebuttal for major comments 3 (temporal breakdown of acquisition steps) and 5 (confusion regarding the axial focusing), they added Supplementary Figure 3. While the figure is informative, it is not sufficient by itself to clear the confusion about the axial focusing:

- The method section should contain the description of the two axial focusing modes with reference to Supplementary Figure 3 (description that already exists in the authors' rebuttal).

We thank the reviewer for this suggestion. We now added the description of the two axial focusing modes with reference to Supplementary Figure 3 under "Imaging speed and data handling" in the methods part. The text now reads: "Thereby, two autofocus variants exist. The free running collective autofocus mode and the focus position map-based triggered mode. In the free running collective autofocus mode the drive speed of the Z-stage is independent of the number of recorded channels. In the focus position map-based triggered mode, each new focus position requires a rest time of ~40 ms (Supplementary Fig. 3)".

- Line 217 should refer to the name of the axial focusing mode used and point to the method for details.

We have added the name of the axial focusing mode used and pointed to the methods and also to supplemental figure 3 for details. The text now reads" To achieve this, the focus position map-based triggered mode was used (see methods and Supplementary Fig.3 for detailed information). Here, the focal plane was calibrated at the beginning of each measurement for all 16 lenses."

- Line 506 should describe what happens to the on-the-fly processing for each axial focusing mode (clarify what happens in the FPGA in each case).

We provide here an extensive explanation for the reviewer below. A shorter version of this text has now been added to the end of the "Imaging speed and data handling" chapter in the methods section of the MS.

The individual imagers do not have a "single shot" mode and will therefore generate images at 30 FPS after system startup, regardless of the selected operating mode of the ComplexEye, with each image being temporarily buffered in the FPGA. This buffering is done according to the "ping-pong" principle, i.e. there are two memories of the same size in the FPGA which are always alternately written with the new image data. Thus, data is always written to one of the two memories while the other still contains a complete previous image. Also, the sharpness indicator generation algorithm is always active in both modes, but has relevance only in the "free running" mode.

What happens now in the "focus position map-based triggered" mode is that the last complete image currently in the FPGA (ping-pong) memory is transferred at trigger time (while at the same time new image data is written to the other of the two memory blocks of the ping-pong memory).

In the "free running" mode, a third memory block comes into play, which is needed to store the last sharpest image of an entire focusing sequence. If the sharpness indicator of a new image is greater than that of the last sharpest image of a focusing sequence, this new image is transferred to the third memory block, thus overwriting the previously sharpest image there. Of course, image data is not copied between the three memory blocks (would take much too long), but the assignment of the function of each of the three memory blocks as "Ping", "Pong" or "Best" memory is done by indexing the blocks with a label of their respective function. This way, the assignment to the three memory functions can be changed in a flash by exchanging the label.

- In the caption of Supplementary Figure 3, the sentence "Autofocus variants of one focusing phase regarding number of video channels and autofocus mode" is complex and not clear to me. Additionally, the name of the two different autofocus modes could be highlighted to improve reading (e.g. quotation marks or bold).

We have re-written the figure caption for more clarity. The caption now reads:

"Supplementary Fig. 3 | **Mode-related focusing speed considerations.** *Two possible autofocus variants, the "free running collective autofocus mode" and the "focus position map-based triggered mode" are shown. While in the "free running collective autofocus mode" the Z-stage is driving permanently in a constant speed and hence the total drive speed is independent of the number of recorded channels (left two diagrams), in the "focus position*

map-based triggered mode” (right two diagrams) each focus level requires a rest time of ~40 ms and the Z-stage pauses at each new position. Here, in conjunction with acceleration and deceleration times of the stage the “focus position map-based triggered mode” exceeds the 8 s limit in a 96 video channel constellation (lower right diagram). Hence, a 96-lens setup would only be possible with the “free-running collective autofocus mode”, if staying in the 8 seconds/frame limit is required.”

Minor:

The authors described a rough breakdown of their cost in the rebuttal (minor comment 7), but while this is interesting for the reviewers, it is an information rather aimed at the readers. I still suggest adding the financial breakdown in the methods of the paper.

I can only assume that the future commercialization of the microscope is the reason for not writing it out directly into the paper. I would then suggest to add something along the line of "This price is a rough estimate that does not take into account price fluctuations, or the development, assembling, and personal costs.". I leave the ultimate decision to include or not this information in the manuscript to the authors.

We appreciate the suggestion of the reviewer and now added a final chapter to the discussion. It reads:

“Although ComplexEye is still a purely experimental system in its current state, we would nevertheless like to provide some rough estimates for its costs. These do not consider price fluctuations or costs for personnel, development or assembly. The procurement costs of all components, semi-finished products and functional assemblies for the 16x system amount to about 50,000 €. The largest items here are the lenses at around €12,000 and the high-precision stage system at €18,000. For a 96x system, which would include 6 of the 16x camera clusters shown in this study, pure hardware costs of about €100,000 can be expected at current prices.”

REVIEWER COMMENTS

Reviewer #1 (Remarks to the Author):

While I maintain the belief that fluorescence remains a pivotal attribute for widespread microscopy system adoption, I concur with the authors in recognizing the innovation and unique applications. Concerning the comparison to the Nikon ECLIPSE system, I still harbor reservations regarding the number of cells tracked per well in the screening experiment. Figure 5d seems to suggest a limited number of tracked cells. The authors provided the number of loaded cells per well in the methods section but did not specify the actual count of tracked cells under each compound treatment. A low number of tracked cells could introduce data noise. It is essential to disclose the number of replicates and the number of tracked cells for each compound in the screening experiments. Furthermore, the Statistical analysis section, particularly concerning the screening experiments, lacks depth. It is advised for the authors to elucidate their criteria for identifying the 17 compound hits and whether false discovery control measures were implemented in the statistical analysis. Additionally, I recommend including all screening data in the supplementary information to facilitate comprehensive understanding for the audience. Once these details are provided, I support the publication of this article in Nature Communications.

Reviewer #2 (Remarks to the Author):

Following the second round of reviews, I consider that the authors have answered all my concerns, and that the manuscript is now much clearer and fit for publication!

Responses to Reviewer 1 for “ComplexEye - a multi lens array microscope for High-Throughput embedded immune cell migration analysis” by Cibir et al.

Reviewer 1:

While I maintain the belief that fluorescence remains a pivotal attribute for widespread microscopy system adoption, I concur with the authors in recognizing the innovation and unique applications. Concerning the comparison to the Nikon ECLIPSE system, I still harbor reservations regarding the number of cells tracked per well in the screening experiment. Figure 5d seems to suggest a limited number of tracked cells. The authors provided the number of loaded cells per well in the methods section but did not specify the actual count of tracked cells under each compound treatment. A low number of tracked cells could introduce data noise. It is essential to disclose the number of replicates and the number of tracked cells for each compound in the screening experiments. Furthermore, the Statistical analysis section, particularly concerning the screening experiments, lacks depth. It is advised for the authors to elucidate their criteria for identifying the 17 compound hits and whether false discovery control measures were implemented in the statistical analysis. Additionally, I recommend including all screening data in the supplementary information to facilitate comprehensive understanding for the audience. Once these details are provided, I support the publication of this article in Nature Communications.

We thank the reviewer for the suggestions.

In Figure 5d, we show all tracks of the related movies normalized to a common starting point (trajectory plot). However, if a compound like the shown R8C4 reduced the migration by 60% and R10G6 even by almost 90%, it is not surprising, that the tracks are very short. This is still presenting several hundred to more than one thousand tracked cells (see details for the compounds in the new EXCEL sheet as described below). For comparison we also show a non-inhibited fMLP experiment, which demonstrates, how much un-inhibited human neutrophils can migrate within one hour in the presence of fMLP.

However, in order to provide exact numbers, we have now taken up the reviewer’s suggestion and created an Excel sheet containing all tracking values for each analyzed compound. These values include:

- Total tracks
- Valid tracks
- Mean speed [$\mu\text{m}/\text{min}$]
- % relative speed
- Activity [%]
- % relative activity

Thereby, valid tracks are determined by two parameters: minimum track duration and movement threshold. For neutrophils, the minimum track duration is defined as 1 min (8 frames or more) and the movement threshold 8 μm (one cell diameter or more). The valid track values of the different compound treatments are in general comparable and show a high number of tracked cells (hundreds to thousands of individual cells), avoiding the risk of data noise.

We have updated the manuscript text and refer to the supplementary Excel table with all the tracking values in the results section. The text now reads: “The detailed tracking results of all tested compounds and the appropriate controls are summarized in the supplementary files (Supplementary Table 1 (Excel sheet)). In total we tracked more than 1.2 million individual neutrophils (mean 1,245 cells per movie) for at least one minute (8 frames) or longer in a total of 1,042 movies. This ensures a very robust data base for our analysis.”

In addition, we have updated the Supplementary Information part and added the following description of the new supplementary Excel sheet: “Supplementary Table 1. Overview of the tracking values for each compound and the respective controls. This Excel sheet shows all evaluated tracking values for the tested compounds and the respective controls (PBS ctr, DMSO ctr, DMSO + fMLP ctr and fMLP ctr, shown in blue colors). Included are the number of total tracks, the number of valid tracks, mean speed in $\mu\text{m}/\text{min}$, % relative speed, activity in % and % relative activity. Thereby, valid tracks are determined by two parameters: minimum track duration and movement threshold. For neutrophils, the minimum track duration is defined as 1 min (8 frames or more) and the movement threshold 8 μm (one cell diameter or more). All movies were recorded for 1 hour with 8 seconds between frames and thus consisted of 450 frames.”

Based on the suggestion of the reviewer, we have also updated the methods section and described in detail the criteria for identifying the 17 hit compounds. Furthermore, we want to mention again that in the first round we did not implement false discovery control measures in the statistical analysis. In the first round, we measured all compounds as singlets. After identifying several hit compounds and as suggested by this reviewer we performed two rounds of validation, which served as control measures for false discovery. The text in the “Methods” section under “Compound screening” now reads: “Thereby, hit compounds were identified based on a threshold. The compound where the cells migrated at least 40% slower than the fMLP control in the same round was defined as a hit. In the first round, all compounds were measured in singlets. After identifying multiple hit compounds, they were validated in two additional rounds.”

REVIEWERS' COMMENTS

Reviewer #1 (Remarks to the Author):

The authors addressed the comments well. I recommend the publication of this manuscript.